# Harnessing metabolic control for synaptic stability: REST/NRSF links glycolytic inhibition to excitatory neurotransmission

Daniele Ferrante[1] [iD], Barbara Parisi[2] [iD], Antonella Marte[1,2], Dionisi Marianna[1], Cosimo Prestigio[1], Andrea Benzi[1], Santina Bruzzone[1,2] [iD], Fabio Benfenati[2,3], Franco Onofri[1,2], Pierluigi Valente[1,2] [iD] and Pietro Baldelli[1,2] [iD]

[1] *Department of Experimental Medicine, University of Genova, Genova, Italy*
[2] *IRCCS Ospedale Policlinico San Martino, Genova, Italy*
[3] *Department of Neuroscience and Brain Technologies, The Italian Institute of Technology IIT, Genova, Italy*

Handling Editors: Katalin Toth & Gareth Morris

The peer review history is available in the Supporting Information section of this article (https://doi.org/10.1113/JP288763#support-information-section).

*The Journal of Physiology*

**Abstract figure legend Schematic of REST/NRSF-dependent postsynaptic downscaling of excitatory strength in response to glycolysis inhibition** Glycolysis inhibition enhances REST/NRSF expression and increases REST/NRSF transcriptional repressor activity by promoting its translocation into the nucleus. Within the nucleus, REST/NRSF specifically downregulates the transcription of the GluA2 gene. Consequently, the decreased expression of postsynaptic glutamate receptors decreases the strength of glutamatergic synapses, with no alterations in the number of active synapses *(Created with* BioRender.com*)*.

D. Ferrante and B. Parisi have contributed as co-first authors.

P. Valente and P. Baldelli have contributed as co-last and co-corresponding authors.

Ethical approval: All experiments were carried out in accordance with the guidelines established by the European Community Council (Directive 2010/63/EU of 4 March, 2014) and were approved by the Italian Ministry of Health (authorization n. 253/2021-PR).

Consent to participate: N/A. The study does not involve any human subjects.

Consent for publication: N/A. The manuscript does not contain any individual persons' data.

**Abstract** Under resting conditions most neuronal ATP is produced through mitochondrial oxidative phosphorylation, whereas glycolysis becomes more important during intense neuronal firing. Recent studies suggest that inhibiting glycolysis plays a key role in regulating seizure-related hyperactivity, with the epigenetic modulator REST/NRSF being activated when glycolysis inhibition lowers the NADH/NAD$^+$ ratio. Our previous research has shown that REST/NRSF initiates homeostatic processes to counteract neuronal hyperactivity by regulating both firing and synaptic activities. However, the exact mechanism through which the metabolic activation of REST/NRSF controls neuronal excitability is still unknown. Here, we studied the role of REST/NRSF in the effects of glycolysis inhibition on hippocampal neuron activity. Treatment with 2-deoxy-D-glucose (2DG) decreased the NADH/NAD$^+$ ratio, increased REST/NRSF expression, and promoted its nuclear translocation. Although GABAergic inhibitory inputs and the firing properties of both excitatory and inhibitory neurons were unaffected by 2DG, the amplitude of evoked EPSCs (eEPSCs) and miniature EPSCs (mEPSCs) was reduced in a REST/NRSF-dependent manner. This effect was associated with a REST/NRSF-dependent reduction in the size of GluA2-positive puncta and a decrease in GluA2 expression in the absence of changes in the density of excitatory synapses. These effects provide a mechanistic basis for the significant reduction in network firing and bursting activity observed when the hippocampal network was treated with 2DG. These findings highlight a role of the REST/NRSF-dependent pathway in the 2DG-mediated downregulation of excitatory inputs, a mechanism that contributes to neuronal network stability, strengthening the homeostatic defences against hyperactivity.

(Received 19 February 2025; accepted after revision 5 September 2025; first published online 1 October 2025)

**Corresponding author** P. Baldelli: Department of Experimental Medicine, University of Genova, Genova, Italy. Email: pietro.baldelli@unige.it

## Key points

- Reducing glucose metabolism with 2-deoxy-D-glucose (2DG) lowers the cell's energy balance and increases the levels of a gene regulator called REST/NRSF.
- REST/NRSF then moves into the nucleus, where it controls the activity of genes linked to nerve cell communication.
- 2DG weakens the strength of signals between excitatory nerve cells, without affecting inhibitory signals or the basic ability of neurons to fire.
- This effect depends in part on REST/NRSF, which reduces the amount and size of GluA2-containing AMPA receptors at excitatory synapses, without altering the overall number of excitatory contacts.
- These findings suggest that blocking glucose metabolism activates a protective response that stabilizes brain networks, which could help control seizures in epilepsy.

**Daniele Ferrante** earned his PhD at the University of Genova, where he also conducted this research as a postdoctoral fellow. He is currently a postdoctoral researcher at the University of Luxembourg, investigating the mechanisms underlying idiopathic and genetic forms of Parkinson's disease. **Barbara Parisi** completed her PhD at the University of Genova on synaptic autophagy and homeostasis, where she also conducted the work presented in this article during her postdoctoral fellowship. She is now a researcher at San Martino Hospital in Genova, focusing on the molecular mechanisms of cognitive dysfunction in schizophrenia.

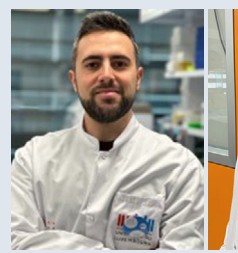
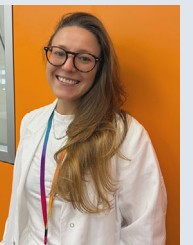

# Introduction

Epilepsy affects over 65 million people worldwide, with approximately one-third of patients being resistant to classical anti-seizure drugs (ASDs). Over the past three decades there has been a growing interest in the use of the high-fat, low-carbohydrate ketogenic diet (KD) and its variants – such as the medium-chain triglyceride (MCT) diet, the modified Atkins diet (MAD), and the low-glycaemic index treatment (LGIT) – as alternative therapies for pharmaco-resistant epilepsy (Rho & Stafstrom, 2012). The common feature of these diets is a significant reduction in carbohydrate intake, leading to decreased glycolysis (Lutas & Yellen, 2013; Rho et al., 2019). This observation, combined with findings that even minimal carbohydrate intake can abolish seizure control achieved by the KD (Huttenlocher, 1976), has spurred a rapid increase in studies evaluating the anti-seizure effects of agents that restrict glycolytic flux, such as 2-deoxy-D-glucose (2DG) (Shao et al., 2018).

2DG is a glucose analogue that prevents the isomerization of glucose-6-phosphate to fructose-6-phosphate, thereby reversibly inhibiting glycolysis. Preclinical studies have demonstrated its significant potential as an anti-seizure agent (Rho et al., 2019). *In vitro* 2DG reduces epileptiform activity across various models, including those induced by high extracellular $K^+$, 4-aminopyridine and bicuculline, as well as in hippocampal slices under GABAergic synaptic activity blockade (Rho et al., 2019; Shao & Stafstrom, 2017; Stafstrom et al., 2009). Its efficacy extends to *in vivo* models, where it has been shown to delay or mitigate seizures under conditions such as kindling, audiogenic seizures and pilocarpine-induced epilepsy (Barton et al., 2001; Gasior et al., 2006; Janicot et al., 2020; Lian et al., 2007; Stafstrom et al., 2009; Yang et al., 2013a). Notably, 2DG is preferentially and rapidly taken up in brain regions with high energy demand, as observed during epileptic seizures.

The mechanism of action of 2DG is complex and multifaceted, involving distinct pathways depending on whether its acute or chronic effects are considered. Acutely, 2DG reduces the frequency and amplitude of spontaneous EPSCs (sEPSCs) when neuronal activity is high, primarily through a presynaptic mechanism (Lujan et al., 2016; Shao & Stafstrom, 2017; Stafstrom et al., 2009). Additionally, 2DG acutely enhances GABAergic signalling by potentiating extrasynaptic tonic GABAergic currents through the activation of neurosteroidogenesis (Forte et al., 2016).

However, the most intriguing effects of 2DG are associated with chronic treatment, which involves transcriptional effects aimed at strengthening homeostatic plasticity processes critical for the long-term control of neuronal network excitability. For instance, 2DG mediates the upregulation of ATP-sensitive $K^+$ ($K_{ATP}$) channel subunits Kir6.1 and Kir6.2 (Lutas et al., 2014; Yang et al., 2013a). These channels, which open when ATP is low due to glycolysis inhibition, promote $K^+$ efflux, leading to cellular hyperpolarization and reduced excitability.

Another compelling mechanism involves the transcriptional repressor REST (RE1-silencing transcription factor), also known as NRSF (neuron-restrictive silencer factor), a master regulator of neuronal homeostasis (Baldelli & Meldolesi, 2015; Lignani et al., 2020). REST/NRSF, along with its co-repressor, the NADH-sensitive carboxy-terminal binding protein (CtBP), is activated by the intracellular decrease in NADH associated with 2DG treatment. Together they target the promoter regions of genes such as *Bdnf* and *TrkB*, reducing seizure progression in the rat kindling model of temporal lobe epilepsy (Garriga-Canut et al., 2006).

In this study, we investigated the involvement of REST/NRSF in the effects exerted by long-term (48 h) treatment with 2DG on the firing properties and synaptic transmission of primary hippocampal neurons. We observed that 2DG promotes REST/NRSF expression and nuclear translocation. Although the firing properties of both excitatory and inhibitory neurons, as well as the strength of inhibitory GABAergic inputs, were unaffected, 2DG induced a downscaling of excitatory glutamatergic transmission that was dependent on REST/NRSF. The underlying mechanism was distinctly postsynaptic and attributed to a specific reduction in the expression of the GluA2 subunit of AMPA-type glutamate receptors.

These findings underscore the ability of glycolysis inhibition to modify the transcriptional profile of neurons in a way that decreases their propensity to hyper-excitability. This is achieved through the recruitment and activation of the transcriptional repressor REST/NRSF, which orchestrates a co-ordinated gene programme that maintains brain circuit activity within physiological limits.

# Results

## Glycolysis inhibition by 2DG does not compromise neuronal survival

Severe hypoglycaemia is known to induce neuronal death (Kleman et al., 2008). Standard culture media for primary neurons contain glucose concentrations (25 mM) that are much higher than physiological levels in the mouse cerebrospinal fluid (1–2 mM). This discrepancy is necessary to sustain neuronal survival *in vitro* due to increased oxidative stress, elevated metabolic demand and limited metabolic support by astrocytes (Kleman et al., 2008). To identify the optimal conditions for inhibiting glycolysis without compromising neuronal viability, we first assessed cell death in hippocampal

neurons maintained for 1 to 4 days in decreased glucose concentrations (25, 10, 5 and 2 mM). No significant increase in cell death was observed ($\sim$ 20%) under these conditions (Fig. 1*A*). Based on this result, we next performed dose–response viability experiments using glucose/2DG combinations with a constant 1:2 ratio

(10/20, 5/10, 2/4 and 1/2 mM), in line with previous studies and theoretical models of competitive inhibition at the levels of glucose transport and hexokinase activity (Garriga-Canut et al., 2006; Harders et al., 2024; Pajak et al., 2020; Zhao et al., 2019). This approach ensures that 2DG effectively competes with glucose for entry

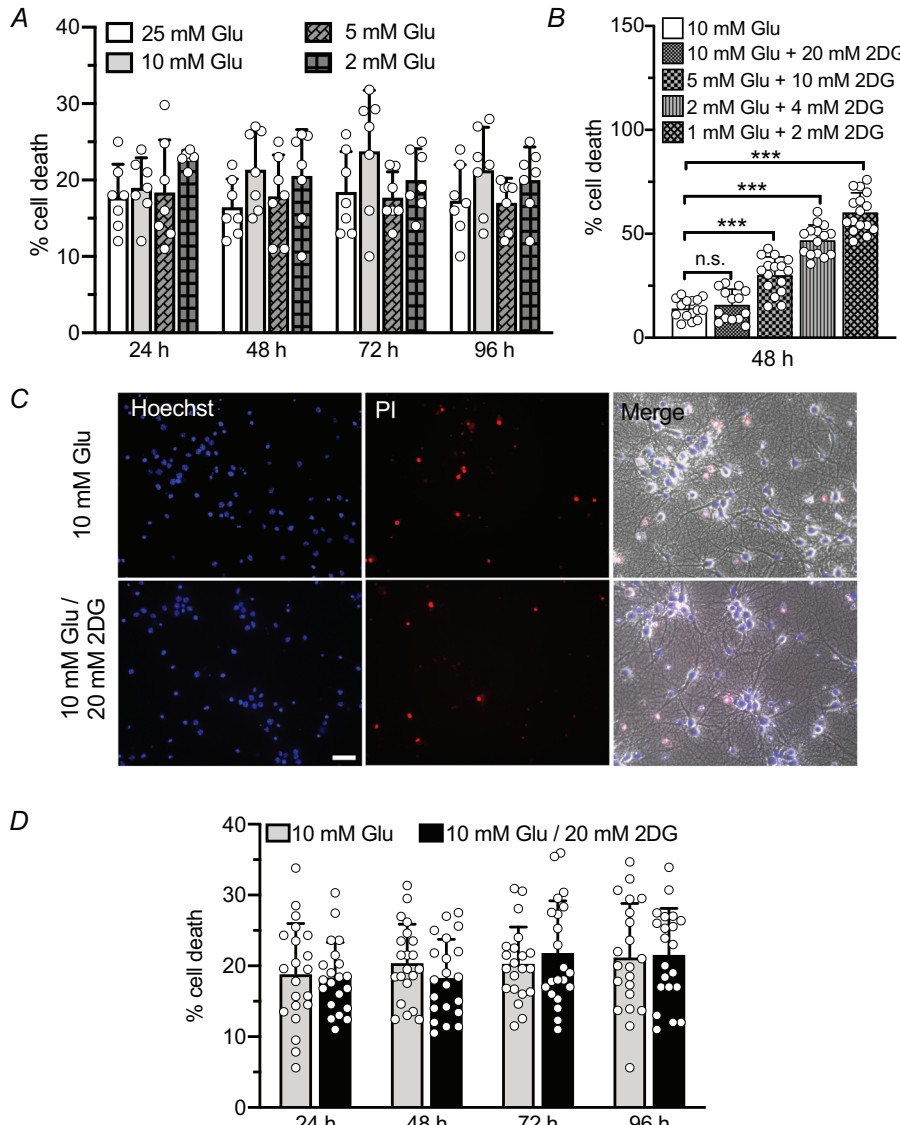

**Figure 1. Cultured hippocampal neurons well tolerate low glucose and 2DG treatments**
*A*, comparison of cell death percentages in cultured hippocampal neurons (12–16 days *in vitro*, div) maintained for 24, 48, 72 and 96 h in culture medium containing 25, 10, 5 or 2 mM glucose. *n* = 7 coverslips from two independent neuronal preparations. Two-way ANOVA showed no significant effect of treatment (*P* = 0.0504) or time (*P* = 0.7895). *B*, bar graph representing neuronal viability (%) following 48 h treatment with glucose to 2DG ratios of 10:20, 5:10, 2:4 and 1:2 mM. *n* = 14–16 fields from two independent neuronal preparations. One-way ANOVA followed by the Dunnett test. 10 Glu *vs*. 10 Glu/20 2DG; *P* = 0.926; for all other comparisons indicated in the figure *P* < 0.0001. *C*, representative fluorescence and differential interference contrast images of cultured hippocampal neurons (14 div) treated for 48 h with 10 mM Glu/vehicle and 10 mM Glu/20 mM 2DG. Scale bar, 20 μm. D-, comparison of cell death percentages in cultured hippocampal neurons (12–16 div) treated for 24, 48, 72 and 96 h with 10 mM Glu/vehicle or 10 mM Glu/20 mM 2DG. *n* = 21 coverslips, three fields per coverslip from three independent neuronal preparations. Two-way ANOVA showed no significant effect of treatment (*P* = 0.8802) or time (*P* = 0.1163). Data are presented as means ± SD with superimposed individual values.

through GLUT3 transporters and inhibits glycolysis by preventing the isomerization of glucose-6-phosphate to fructose-6-phosphate (Laussel & Léon, 2020). Our goal was to identify the glucose/2DG combination that inhibits glycolysis without inducing neuronal toxicity after 48 h of treatment. We found that neuronal viability was optimal at 10/20 mM, whereas glucose/2DG combinations at lower glucose concentrations led to progressively increased neuronal death (Fig. 1*B*). Finally we tested the effect of the 10/20 mM glucose/2DG combination over prolonged treatment periods (24, 48, 72 and 96 h) and found no significant reduction in neuronal viability even after sustained exposure (Fig. 1*C, D*). Therefore we used 20 mM 2DG throughout to minimize confounding effects due to stress or cytotoxicity (Kaneko et al., 2014; Noh et al., 2012; Palm et al., 1998).

## Glycolysis inhibition reduces the NADH/NAD$^+$ ratio and induces the translocation of REST/NRSF into the nucleus

Glycolysis is a key metabolic pathway that generates ATP and NADH, with NADH being an important electron carrier in cellular metabolism (Magistretti & Allaman, 2015). When neuronal glycolysis was inhibited by 2DG treatment for 2 h, the level of NADH relative to NAD$^+$ was reduced (Fig. 2*A*), resulting in a significant decrease in the NADH/NAD$^+$ ratio (Fig. 2*B*). Glycolysis-derived NADH is known to be an allosteric regulator of the REST/NRSF co-repressor CtBP, modulating its interaction with REST/NRSF (Garriga-Canut et al., 2006; Zhang et al., 2002). Thus, CtBP could act as a redox sensor that directly integrates metabolic demands with gene expression. To study the transcriptional activity of REST/NRSF, primary hippocampal neurons were treated with a Cy3-tagged decoy oligodeoxynucleotide, complementary to the RE1-binding domain of REST (Cy3-ODN) that was chemically modified to improve its stability (Soldati et al., 2011) and used to trace endogenous REST/NRSF. A random decoy oligodeoxynucleotide sequence (Cy3-NEG) served as a negative control. Both Cy3-ODN and Cy3-NEG, added extracellularly, were able to cross the plasma membrane and diffuse into the cytosol efficiently (Fig. 2*C*). Interestingly, upon 4 h treatment with 2DG Cy3-ODN doubled its partitioning ratio into the nucleus, whereas the cytosolic distribution of Cy3-NEG was unaffected (Fig. 2*D,E*). These results suggest that the inhibition of glycolysis activates REST/NRSF translocation from the cytoplasm to the nucleus.

## Glycolysis inhibition by 2DG upregulates REST/NRSF

It has been demonstrated that neuronal REST/NRSF levels, which are low under physiological conditions, elevate in response to stress-inducing stimuli, such as seizures, ischaemia or neuroinflammation (Buffolo et al., 2021; Kaneko et al., 2014; Palm et al., 1998). Treatment of cultured hippocampal neurons with 2DG for 6 h resulted in a significant increase in *Rest/Nrsf* mRNA levels, and this effect persisted for up to 12 h (Fig. 3*A*). Additionally, it has been shown that glycolysis inhibition reduces the upregulation of BDNF observed in the hippocampus of a rat kindling model (Garriga-Canut et al., 2006). Therefore we investigated whether treatment with 2DG could alter *Bdnf* transcription under our *in vitro* conditions. We observed that 2DG had no effect on the transcription of this neurotrophin (Fig. 3*B*), which is known to play a crucial role in the process of epileptogenesis (Binder et al., 2001). In contrast, 2DG treatment for 24 h led to an increase in REST/NRSF protein expression (Fig. 3*C*), consistent with the increase in mRNA levels observed at shorter treatment times (6 and 12 h).

## 2DG does not affect the firing activity of excitatory and inhibitory neurons

As Na$^+$ channels are major neuronal REST/NRSF targets (Chong et al., 1995; Schoenherr & Anderson, 1995) and are downregulated by REST/NRSF (Nadeau & Lester, 2002; Pozzi et al., 2013), we hypothesized that 2DG treatment might affect neuronal firing properties through REST/NRSF activation. To test this possibility, the effect of 48 h treatment with 2DG on firing activity was studied after REST/NRSF inhibition by oligodeoxynucleotides (ODNs) that sequester the transcription factor from its genomic binding sites (Prestigio et al., 2021; Soldati et al., 2011). Patch-clamp recordings in the current-clamp configuration were used to deliver constant current pulses of increasing amplitude and measure the resulting action potential (AP) firing in cultured hippocampal neurons obtained from GAD67-GFP mice (Tamamaki et al., 2003). As previously shown GFP-negative glutamatergic excitatory neurons showed lower firing frequency than GFP-positive GABAergic interneurons (Fig. 4*A*) (Prestigio et al., 2019). The maximum instantaneous firing frequency, in response to a 250 pA current injection, showed that neither the 48 h treatment with 2DG nor with 2DG + ODN affected the AP discharge frequency in both excitatory and inhibitory neurons (Fig. 4*B*). Identical results were obtained when studying the effect of 2DG and ODN on the half-width and peak amplitude of single APs (Fig. 4*C, D*), as well as on all other parameters describing the AP waveform and the passive membrane properties of excitatory and inhibitory neurons (Table 1). In accordance with the complete lack of effect of 2DG or REST/NRSF inhibition immunoblotting analysis showed no significant changes in the expression levels of voltage-gated Na$^+$ channels (Na$_V$) (Fig. 4*E*). These data indicate that the inhibition of neuronal glycolysis does not induce any

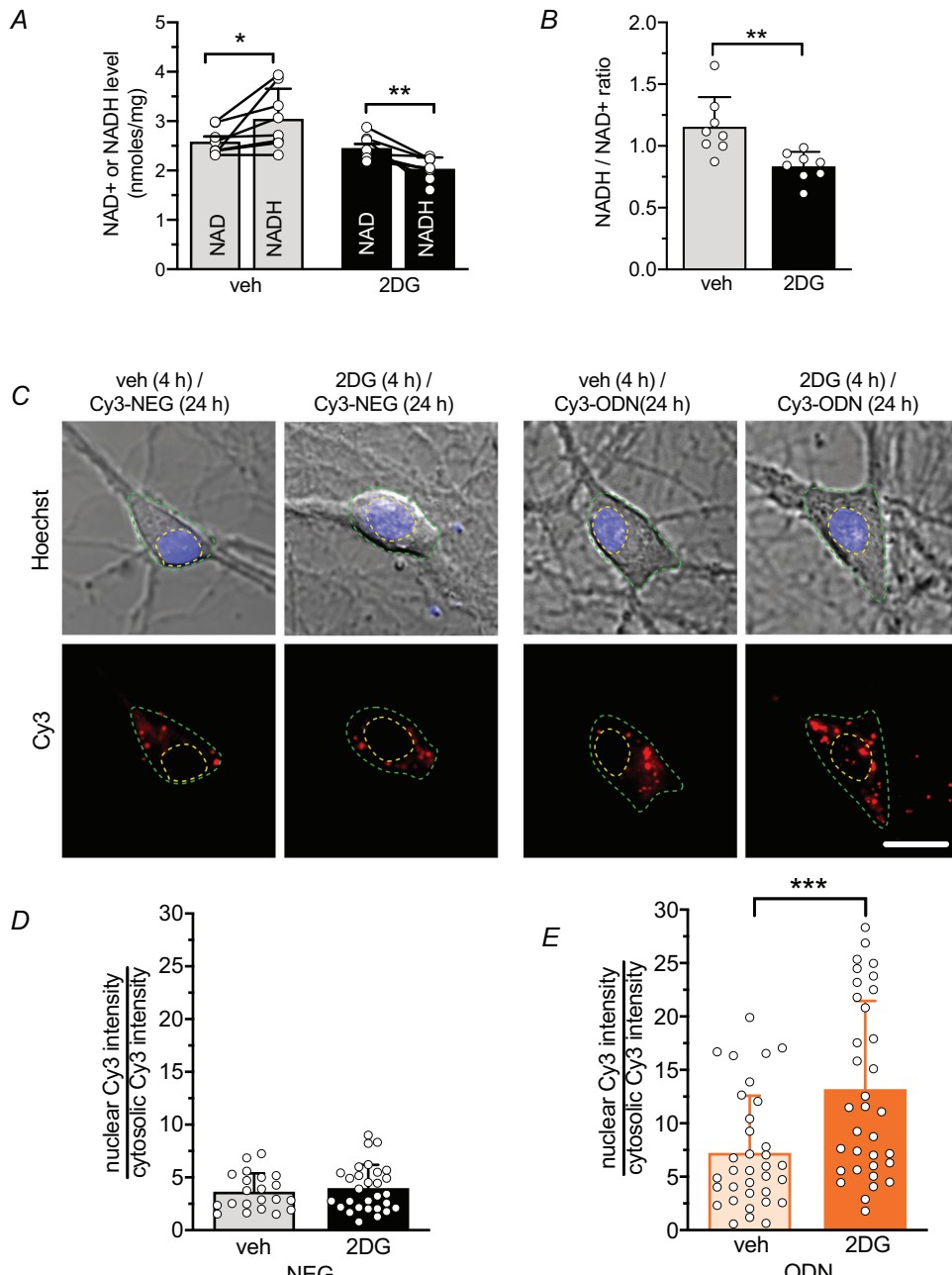

**Figure 2. 2DG treatment decreases NADH/NAD⁺ ratio and favours the translocation of REST/NRSF to the nucleus**

*A,B*, quantitative analysis (means ± SD) of NAD⁺ and NADH levels (*A*) and NADH/NAD+ ratio (*B*) in cultured hippocampal neurons (14 div) treated for 2 h with 2DG or vehicle. *n* = 8 samples from two independent neuronal preparations. *A*, paired Student's *t* test, *P* = 0.041 veh; *P* = 0.007 2DG. *B*, unpaired Student's *t* test, *P* = 0.0044. *C*, representative differential interference contrast (*upper panels*) and fluorescence images (*lower panels*) showing cultured hippocampal neurons (14 div) treated with 2DG or vehicle for 4 h and subsequently labelled with either Cy3-tagged ODN (Cy3-ODN) or Cy3-tagged NEG (Cy3-NEG) for 24 h. Scale bar, 10 μm. Tracking of Cy3-NEG (*D*) and Cy3-ODN (*E*) reveals both cytoplasmic and nuclear localizations of REST/NRSF and its nuclear translocation upon 2DG treatment. Bar plots represent the means ± SD with superimposed individual values. *n* = 21–34 fields from three independent preparations. *D*, unpaired Student's *t* test, *P* = 0.546; *E*, Mann–Whitney *U* test, *P* = 0.0009.

**Table 1. Passive and active membrane properties of excitatory and inhibitory cultured hippocampal neurons treated with NEG/veh, NEG/2DG, ODN/veh and ODN/2DG for 48 h**

| | Glutamatergic neurons | | | | | GABAergic neurons | | | | |
| | NEG | | ODN | | | NEG | | ODN | | |
| | veh | 2DG | veh | 2DG | P | veh | 2DG | veh | 2DG | P |
|---|---|---|---|---|---|---|---|---|---|---|
| Membrane capacitance (pF) | 59.72 ± 17.35 | 51.41 ± 18.22 | 62.78 ± 15.9 | 52.48 ± 17.30 | 0.465 | 54.27 ± 12.54 | 50.95 ± 12.61 | 61.67 ± 17.60 | 56.27 ± 14.26 | 0.976 |
| Resting potential (mV) | −57 ± 10.47 | −59 ± 5.38 | −59.7 ± 7.498 | −61.1 ± 6.41 | 0.972 | −61.09 ± 7.32 | −59.44 ± 6.04 | −60.67 ± 9.24 | −57.22 ± 11.85 | 0.998 |
| AP threshold (mV) | −35,37 ± 6.411 | −31,81 ± 4.58 | −31,89 ± 6.03 | −32,85 ± 4.29 | 0.282 | −34.78 ± 3.22 | −32.34 ± 2.81 | −33.9 ± 2.98 | −34.01 ± 2.97 | 0.406 |
| AP amplitude (mV) | 50.98 ± 6.65 | 49.79 ± 8.18 | 47.64 ± 1.478 | 49.22 ± 1.302 | 0.996 | 86.12 ± 5.3535 | 84.44 ± 5.381 | 83.83 ± 5.610 | 85.8 ± 5.208 | 0.976 |
| Half-width (ms) | 2.51 ± 0.48 | 2.433 ± 0.33 | 2.294 ± 0.48 | 2.4 ± 0.50 | 0.997 | 1.365 ± 0.18 | 1.338 ± 0.26 | 1.151 ± 0.15 | 1.143 ± 0.26 | 0.999 |
| Afterhyperpolarization (mV) | −49.06 ± 6.15 | −49.96 ± 8.18 | −42.25 ± 4.21 | −45.37 ± 7.20 | 0.923 | −46.64 ± 3.91 | −46.81 ± 4.54 | −47.02 ± 4.31 | −48.51 ± 5.85 | 0.999 |
| Max rising slope (mV/ms) | 194.2 ± 71.26 | 163.9 ± 60.28 | 196.8 ± 74.93 | 165.4 ± 61.47 | 0.832 | 452.9 ± 109.9 | 396 ± 112.8 | 446.3 ± 94.76 | 427.6 ± 102.5 | 0.721 |
| Max repolarization slope (mV/ms) | 37.92 ± 11.86 | 34.25 ± 11.23 | 35 ± 8.50 | 34.92 ± 10.83 | 0.955 | 95.83 ± 30.08 | 95.08 ± 24.52 | 93.85 ± 17.48 | 83.67 ± 18.18 | 0.999 |
| Instantaneous firing frequency (Hz) | 19.42 ± 4.81 | 20.22 ± 4.55 | 20.4 ± 4.54 | 20.53 ± 4.17 | 0.949 | 48.13 ± 9.82 | 47.81 ± 10.57 | 51.38 ± 10.46 | 51,88 ± 12.18 | 0.990 |

*Note*: Data are presented as means ± SD. Sample size: $n = 7$–23 wells per group from three independent neuronal preparations. Statistical analysis was performed using two-way ANOVA followed by the Tukey test. Reported *P*-values refer to comparisons between NEG/vehicle and NEG/2DG conditions.

changes in neuronal excitability in both inhibitory and excitatory neurons.

### Glycolysis inhibition by 2DG induces a REST/NRSF-dependent downscaling in excitatory synaptic strength

Hippocampal neurons grown as autaptic cells enable the investigation of changes in synaptic strength and quantal parameters of synaptic transmission from a homogeneous population of synapses (Valente et al., 2016b). Thus, patch-clamp recordings were used to examine the impact of glycolysis inhibition on evoked eEPSCs or eIPSCs in autaptic neurons obtained from GAD67-GFP mice (12–15 days *in vitro*, div). In GFP-negative excitatory autaptic neurons (Fig. 5*A*, *top*), 48 h treatment with 2DG halved eEPSC amplitudes, an effect that was fully prevented by co-treatment with ODN (Fig. 5*B,C*). This reduction occurred without significant changes in the paired-pulse ratio (PPR; Fig. 5*B,C*), suggesting a postsynaptic mechanism (Fioravante & Regehr, 2011). Conversely GFP-positive inhibitory autaptic neurons (Fig. 5*A*, *bottom)* were insensitive to 2DG, with no detectable changes in eIPSC amplitude and PPR (Fig. 5*D*). These findings indicate that 2DG selectively inhibits the excitatory synaptic strength while fully preserving inhibitory synaptic transmission. This selective action highlights the capacity of 2DG to reduce the excitatory/inhibitory (E/I) balance through a specific modulation of excitatory synapses.

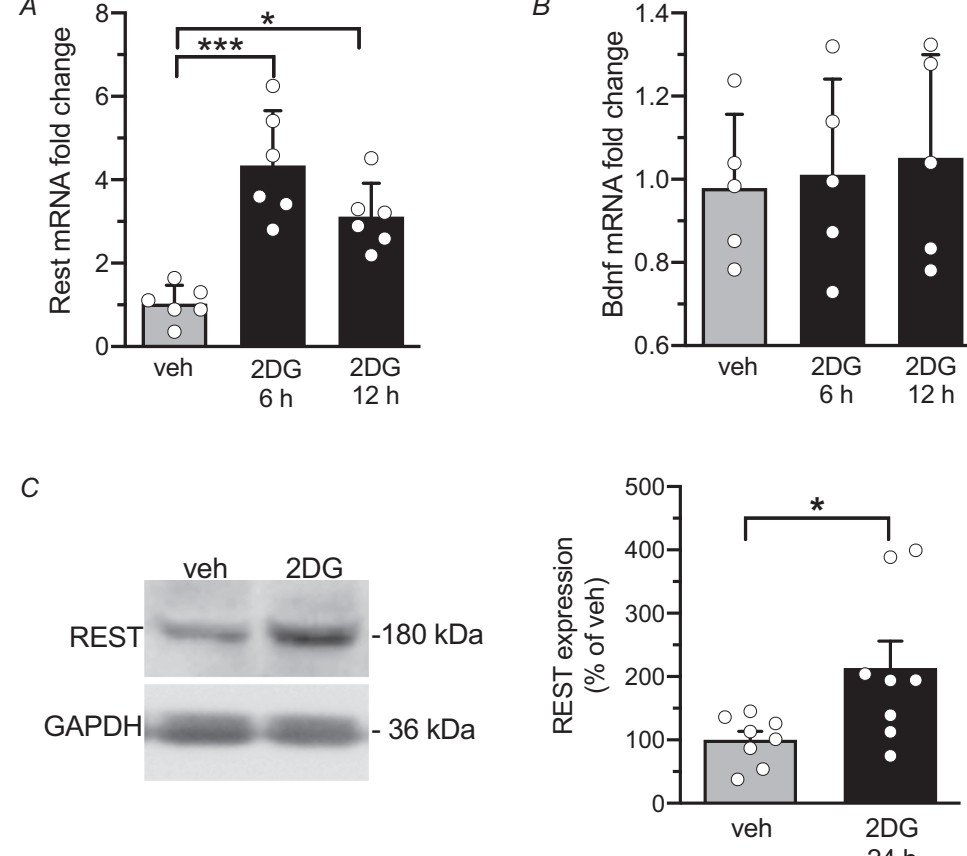

**Figure 3. 2DG treatment increases REST/NRSF mRNA and protein without affecting BDNF expression**
*A,B*, time course of *Rest/Nrsf (A)* and *Bdnf (B)* mRNA fold changes in primary cortical neurons (12 div) treated with vehicle (*grey*) and 20 mM 2DG (black) for 6 and 12 h. Data represent means ± SD with superimposed individual values. *A*, *n* = 6 samples from two independent neuronal preparations. veh *vs*. 2DG 6 h *P* = 0.0007; veh *vs*. 2DG 12 h *P* = 0.0463. Kruskal-Wallis ANOVA followed by the Dunn's test. *B*, *n* = 5 samples from two independent neuronal preparations. veh *vs*. 2DG 6 h *P* = 0.99; veh *vs*. 2DG 12 h *P* = 0.99. Kruskal-Wallis ANOVA followed by the Dunn's test. *C, Left*: representative immunoblots showing REST/NRSF protein levels in total cortical lysates (13 div) treated for 24 h with either vehicle or 2DG. GAPDH immunoreactivity was used as a control for equal loading. *Right*: Quantification of REST/NRSF protein expression was normalized to GAPDH expression and represented in percentage of the mean vehicle immunoreactivity. Data are presented as means ± SD with superimposed individual values; *n* = 8 from three independent preparations; unpaired Student's *t* test, *P* = 0.0234.

## 2DG induces a REST/NRSF-dependent reduction in the readily releasable pool size without affecting the release probability in excitatory synapses

To better define the mechanism by which glycolysis inhibition modulates glutamate release, we estimated the readily releasable pool for synchronous release ($RRP_{syn}$) and the probability of release (Pr) using the cumulative amplitude analysis. When excitatory autapses were challenged with a 40 Hz stimulation train lasting 2 seconds (80 APs), a significant depression of eEPSCs became evident during the stimulation period regardless

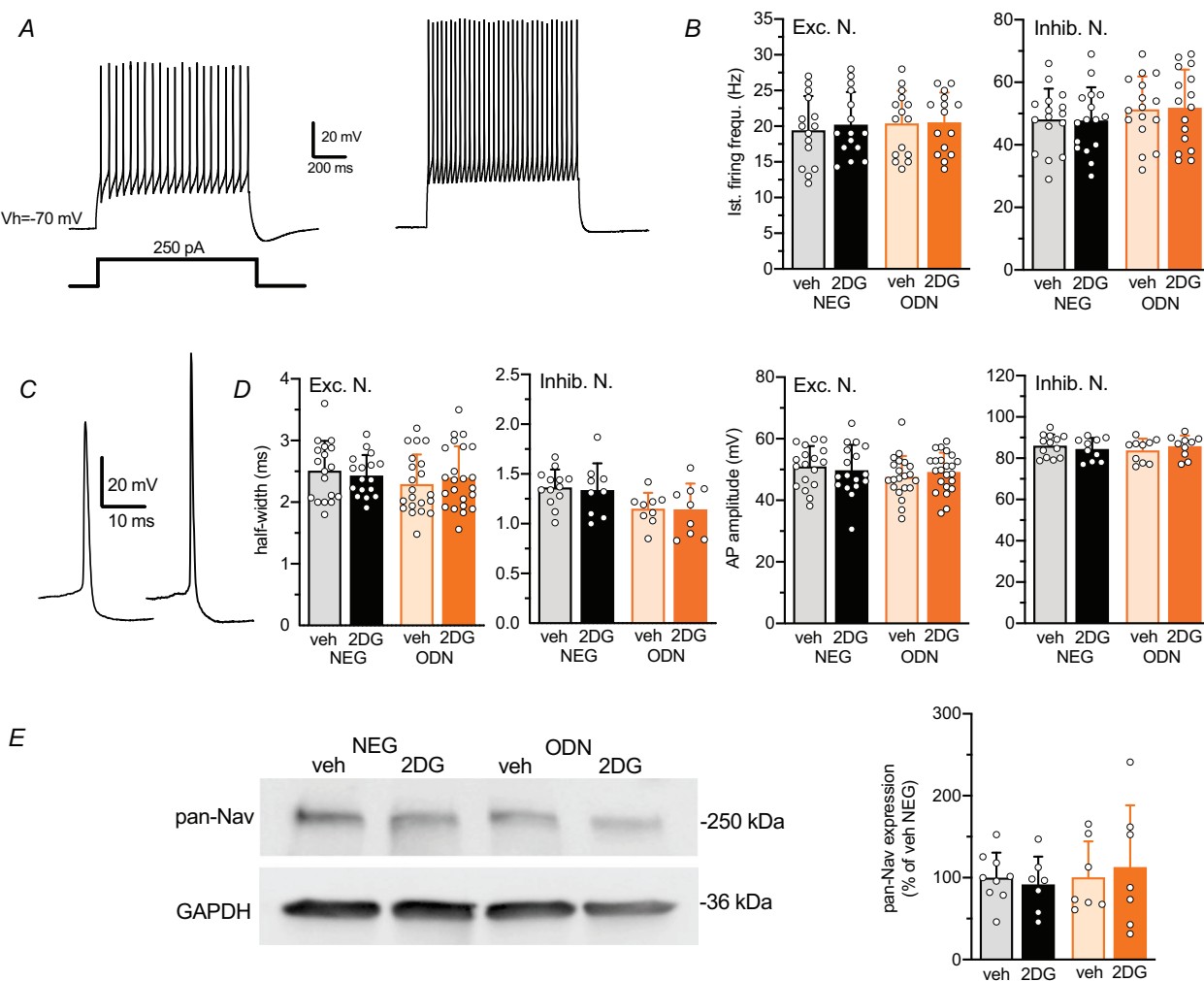

**Figure 4. 2DG treatment does not affect the firing properties of both glutamatergic and GABAergic neurons**

*A*, representative whole-cell current-clamp recordings of APs evoked by injecting 1 s current step of 250 pA in glutamatergic (*left*) and GABAergic (*right*) neurons. *B*, bar plots show the means ± SD with superimposed individual values of the instantaneous firing frequency in excitatory (*left*) and inhibitory (*right*) neurons treated with NEG/veh, NEG/2DG, ODN/veh or ODN/2DG for 48 h. *n* = 15–16 from three independent neuronal preparations. Two-way ANOVA, *P* = 0.8786 and *P* = 0.15 for excitatory and inhibitory neurons, respectively. *C*, representative current-clamp recordings showing the shape of the first AP elicited at the rheobase in glutamatergic (*left*) and GABAergic (*right*) neurons. *D*, bar graphs show the means ± SD with superimposed individual values of the half-width and amplitude of APs recorded in excitatory (*left*) and inhibitory (*right*) neurons treated with NEG/veh, NEG/2DG, ODN/veh or ODN/2DG for 48 h. *n* = 9–23 from three independent neuronal preparations. Two-way ANOVA. For half-width: *P* = 0.891 and *P* = 0.801 for excitatory and inhibitory neurons, respectively. For AP amplitude *P* = 0.901 and *P* = 0.929 for excitatory and inhibitory neurons, respectively. *E*, *left*: representative immunoblots of pan-Na$_V$ immunoreactivity in primary cortical neurons (12 div) treated as described above. GAPDH immunoreactivity was included as a control for equal loading. *Right*: Quantification of Na$_V$ channel immunoreactivity expressed in percentage of the NEG/veh control. Data are presented as means ± SD with superimposed individual values. *n* = 9–7 from three independent preparations. Two-way ANOVA, *P* = 0.905.

of the amplitude of the first current in the train (Fig. 6*A*). We analysed the cumulative profile of eEPSC amplitude, which displayed a rapid initial rise followed by a slower linear increase during later pulses (Fig. 6*B*). Assuming that the slow linear increase is attributable to the equilibrium between release and the constant replenishment of synaptic vesicles, back-extrapolation of the linear portion to time 0 for individual neurons yields the total release minus the total replenishment corresponding to the $RRP_{syn}$ (Schneggenburger & Neher,

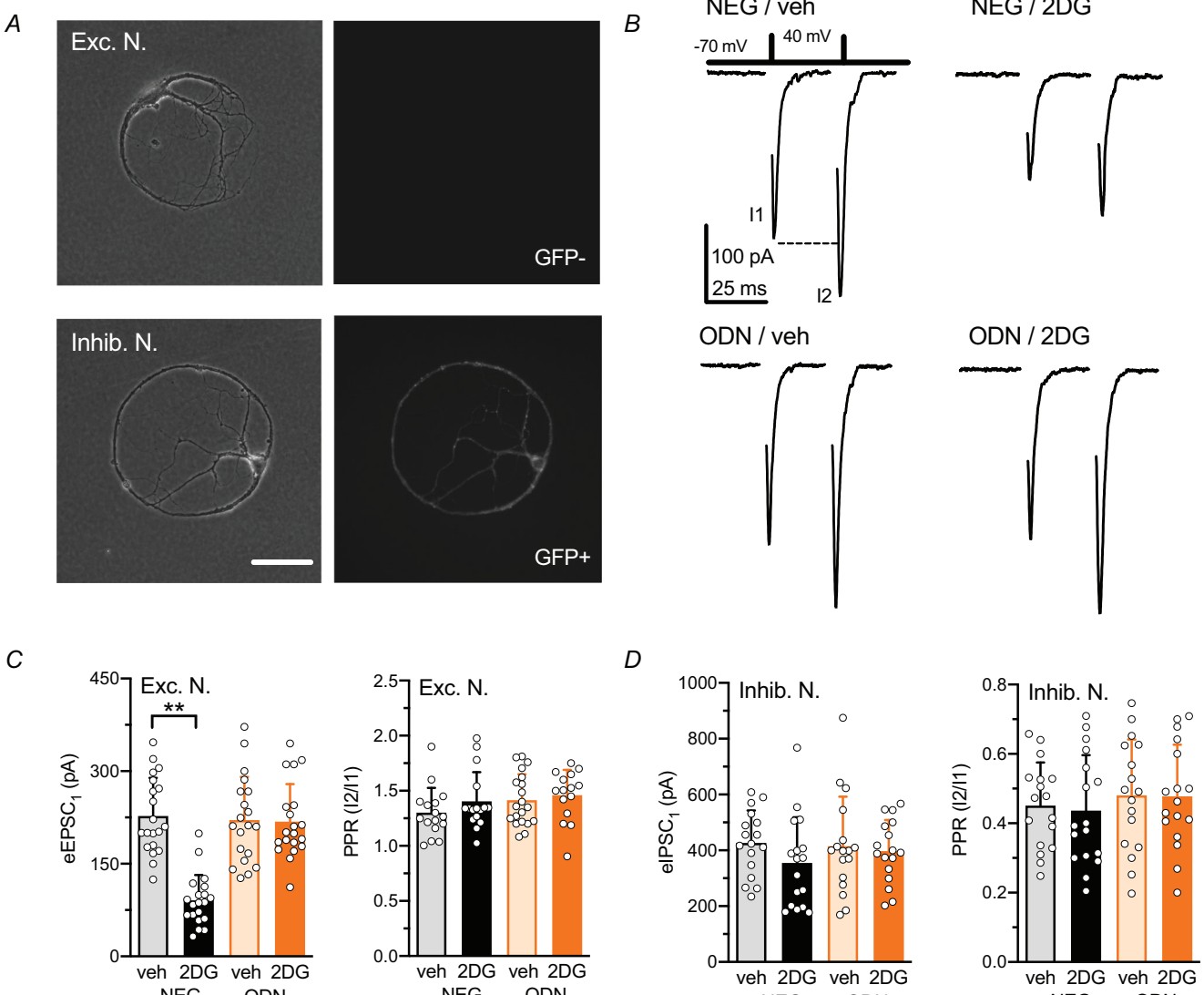

**Figure 5. Excitatory, but not inhibitory, synaptic strength is decreased by 2DG treatment in a REST/NRSF-dependent manner**

*A*, phase-contrast micrographs of a glutamatergic (GAD67-GFP negative, *top*) and GABAergic (GAD67-GFP positive, *bottom*) hippocampal autaptic neurons. Scale bar, 50 μm. *B*, representative eEPSCs recorded from excitatory GAD67-GFP negative autaptic neuron treated with NEG or ODN in the presence of either vehicle or 2DG. eEPSCs were elicited by clamping the neuron at −70 mV and applying two voltage steps (+40 mV, 0.5 ms) with an inter-stimulus interval of 50 ms. Paired-pulse stimulation was applied every 10 s. Stimulation artifacts were removed for clarity. *C*, bar graphs showing the mean ± SD (with superimposed individual values) eEPSC amplitude and paired-pulse ratio (PPR) from excitatory neurons treated with NEG/veh, NEG/2DG, ODN/veh, or ODN/2DG for 48 h. *n* = 16–20 neurons from 3 independent preparations. Two-way ANOVA followed by the Tukey's test. For eEPSCs: NEG/veh *vs*. NEG/2DG $P < 0.0001$. For PPR: NEG/veh *vs*. NEG/2DG $P = 0.648$. *D*, bar graphs showing the mean ± SD (with superimposed individual values) eIPSC amplitude and paired-pulse ratio (PPR) from inhibitory neurons treated with NEG/veh, NEG/2DG, ODN/veh, or ODN/2DG for 48 h. *n* = 16–17 neurons from 3 independent preparations. Two-way ANOVA followed by the Tukey's test. For eIPSCs amplitude: NEG/veh *vs*. NEG/2DG $P = 0.475$. For PPR: NEG/veh *vs*. NEG/2DG $P = 0.991$.

2000). In NEG-treated neurons 2DG slowed the initial rise of the cumulative current, which reached significantly lower plateau levels, thereby reducing the mean $RRP_{syn}$ (Fig. 6C, *left*). This effect of 2DG was virtually abolished in ODN-treated neurons (Fig. 6C, *left*). In contrast, the Pr, calculated as the ratio between the first EPSC in the train ($I_1$) and $RRP_{syn}$ (see Materials and Methods), was not affected by 2DG in either NEG-treated or ODN-treated neurons (Fig. 6C, *right*). These data demonstrate that the 2DG-induced reduction in eEPSC amplitude is due to a decrease in $RRP_{syn}$.

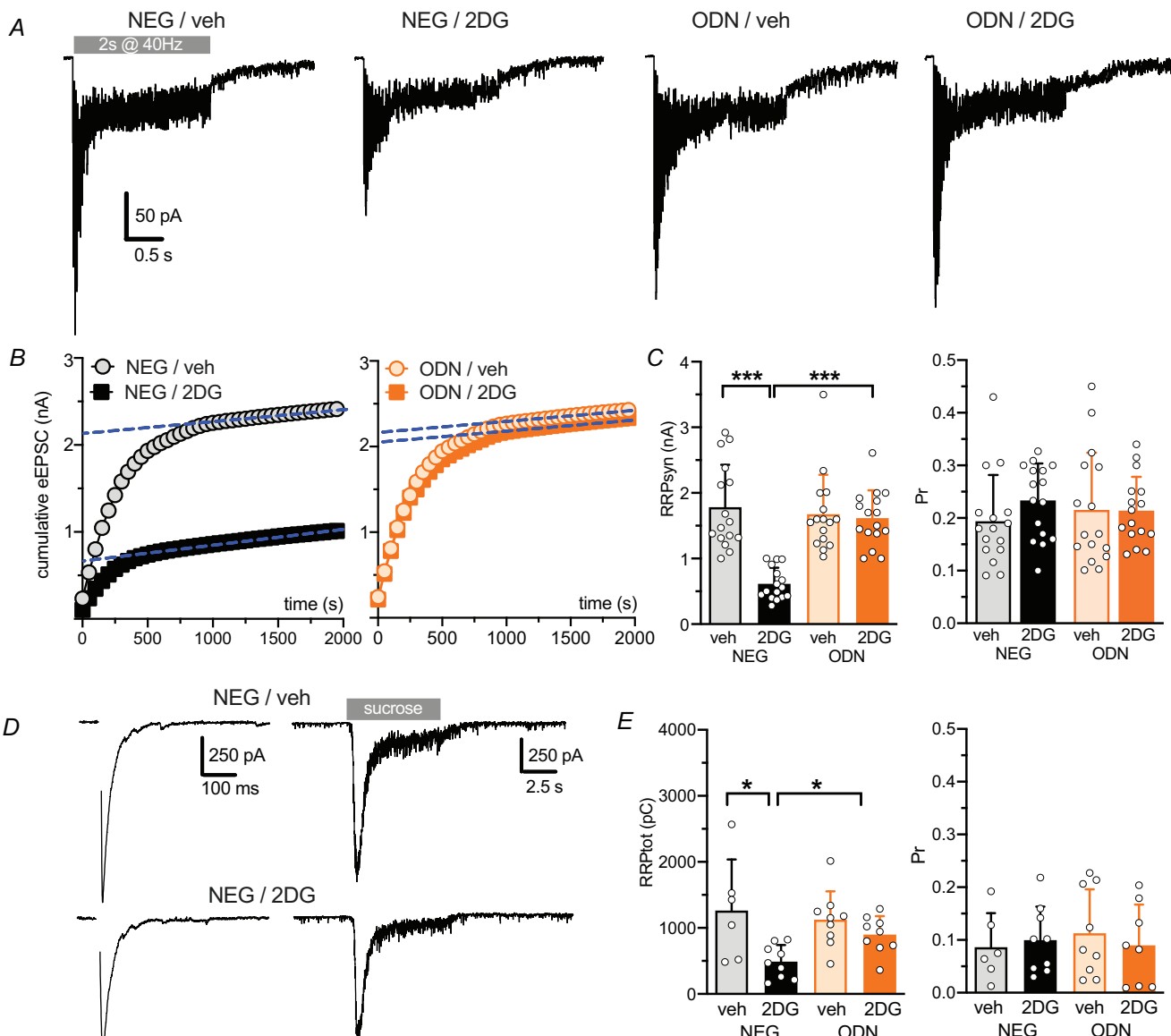

**Figure 6. 2DG treatment decreases the size of the readily releasable pool of excitatory synapses**

*A*, representative recordings of eEPSCs evoked by a brief tetanic stimulus (2 s at 40 Hz) in excitatory autaptic neurons treated with NEG/veh, NEG/2DG, ODN/veh, or ODN/2DG for 48 h. Stimulation artifacts were removed from each trace for clarity. *B*, representative cumulative amplitude profiles of eEPSCs recorded as in (*A*). *C*, bar plots showing the mean ± SD (with superimposed individual values) $RRP_{syn}$ (*left*) and Pr (*right*) from glutamatergic autaptic neurons treated as in (*A*). n = 16–17 neurons from three independent preparations. Two-way ANOVA followed by the Tukey test. For $RRP_{syn}$: NEG/veh *vs*. NEG/2DG, $P < 0.0001$. For Pr: NEG/veh *vs*. NEG/2DG, $P = 0.475$. *D*, representative eEPSCs and responses to hypertonic solution for NEG/veh and NEG/2DG neurons. *E*, bar plots showing the mean ± SD (with superimposed individual values) $RRP_{tot}$ (*left*) and Pr (*right*) from glutamatergic low-density neurons treated as in (*A*). n = 6–9 neurons from two independent preparations. Two-way ANOVA followed by the Tukey test. For $RRP_{tot}$: NEG/veh *vs*. NEG/2DG, $P = 0.036$; NEG/2DG *vs*. ODN/2DG, $P = 0.012$. For Pr: NEG/veh *vs*. NEG/2DG, $P = 0.999$; NEG/2DG *vs*. ODN/2DG, $P = 0.999$.

The RRP and the Pr were also estimated using hypertonic stimulation (Rosenmund & Stevens, 1996). Afferent fibres to a voltage-clamped hippocampal neuron were initially stimulated extracellularly at low frequency (0.05 Hz) to evoke maximal eEPSCs. One minute later, a hypertonic solution was focally applied to the same neuron. As previously described (Baldelli et al., 2007; Moulder & Mennerick, 2005), the charge transfer of the transient current induced by high-sucrose application – an established measure of total RRP ($RRP_{tot}$) – was significantly reduced in NEG/2DG-treated neurons. In contrast no significant difference was observed in the Pr, calculated as the ratio between the eEPSC charge and $RRP_{tot}$ (Fig. 6D,E).

### Inhibition of glycolysis by 2DG results in a REST/NRSF-dependent decrease in mEPSC amplitude

The reduction in RRP observed through the cumulative amplitude analysis and hyperosmotic stimulation method can be interpreted as a decrease in the number of synaptic vesicles available for release (*quantal content*) or a decrease in the current generated by the fusion of a single vesicle (*quantal size*). The study of miniature EPSCs (mEPSCs) is undoubtedly the most effective method to functionally evaluate these aspects. Specifically, the amplitude of mEPSCs reflects the postsynaptic effects of neurotransmitters and depends on the number of glutamatergic receptors present on the postsynaptic membrane. In contrast mEPSC frequency is mostly dependent on the number of active excitatory synapses contacting the patched neuron (Fioravante & Regehr, 2011). mEPSCs were continuously recorded at the soma of a voltage-clamped neuron held at −70 mV and locally perfused with a Tyrode solution containing TTX (1 μM) and bicuculline (30 μM) to block spontaneous APs and GABAergic postsynaptic currents, respectively (Fig. 7A). 2DG treatment significantly reduced mEPSC amplitude, an effect that was entirely suppressed in neurons where REST/NRSF activity was blocked by ODN (Fig. 7A,B).

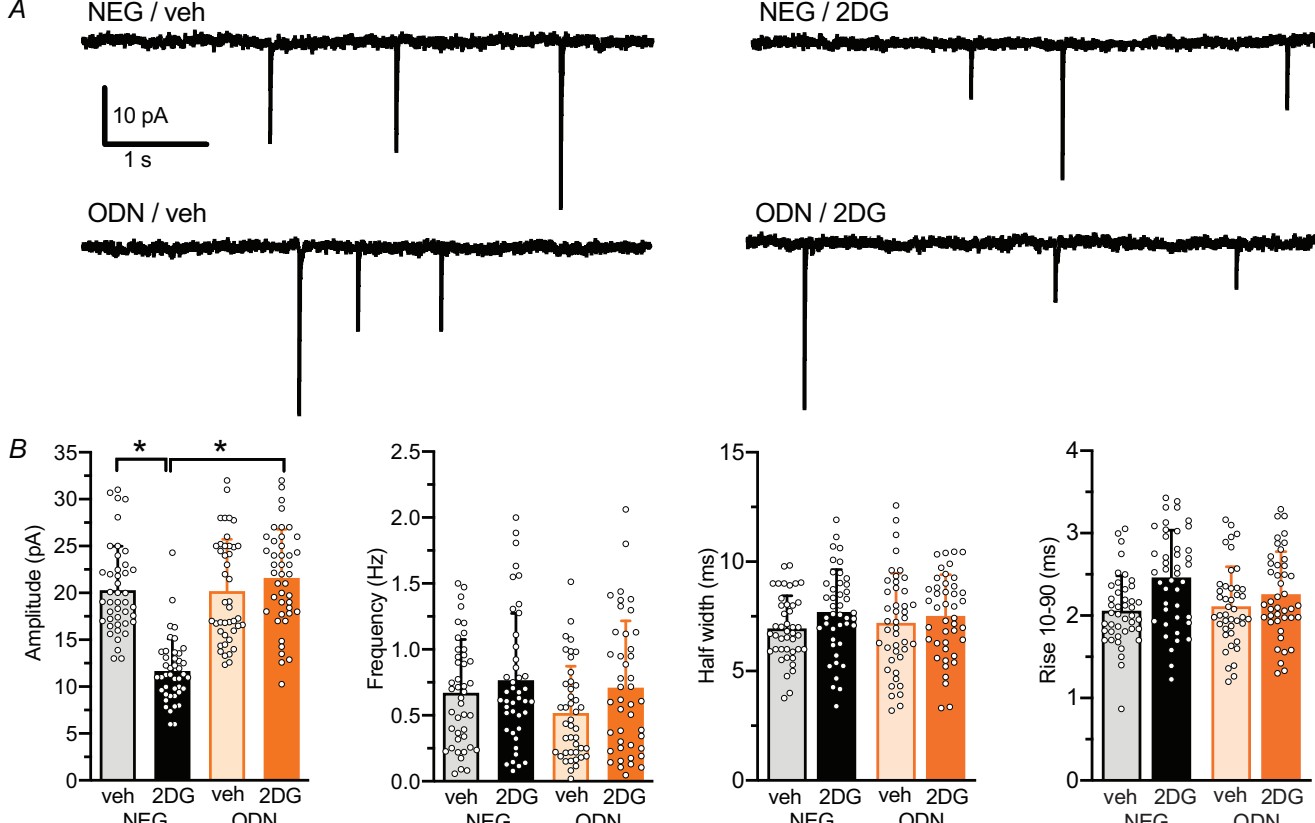

**Figure 7. 2DG treatment induces REST/NRSF-dependent downregulation of the amplitude of miniature EPSCs**
*A*, representative traces of mEPSCs recorded in low-density hippocampal neurons (14–15 div) treated with NEG/veh, NEG/2DG, ODN/veh or ODN/2DG for 48 h. *B*, mean ± SD with superimposed individual values of mEPSC amplitude, frequency, half-width and 10%–90% rise time. *n* = 43 neurons from five independent preparations. Two-way ANOVA followed by the Tukey test. NEG/veh *vs*. NEG/2DG, $P < 0.0001$, $P = 0.748$, $P = 0.264$ and $P = 0.0613$ for mEPSC amplitude, frequency, half-width and 10%–90% rise time, respectively.

Conversely mEPSC frequency and kinetics remained comparable between untreated and 2DG-treated neurons (Fig. 7*B*). These findings suggest that glycolysis inhibition induces a REST/NRSF-dependent reduction in the expression of glutamatergic receptors in the absence of changes in the number of active synaptic contacts.

### Glycolysis inhibition by 2DG reduces the expression of the GluA2 subunit of AMPA glutamate receptors

To confirm the lack of any effect of 2DG on the number of excitatory glutamatergic synapses at the soma or in more distal dendritic regions, we performed double immuno-staining of hippocampal neurons using antibodies against Homer and VGLUT1 to label the excitatory postsynaptic scaffold and presynaptic terminal, respectively (Fig. 8*A*). The morphometric analysis of the distribution of the double-labelled excitatory synapses confirmed that the 2DG treatment did not affect the density of excitatory synapses at the cell body and the distal dendritic levels (Fig. 8*B*).

As activation of REST/NRSF by various neuronal insults was reported to affect the expression of the GluA2 subunit of AMPA-type glutamate receptors (AMPARs) (Hwang & Zukin, 2018), we repeated the double staining of primary hippocampal neurons using antibodies to VGLUT1 and GluA2 (Fig. 8*C*). The analysis of excitatory synaptic puncta revealed a significant reduction in the integrated fluorescence intensity of GluA2 following glycolysis inhibition by 2DG. This effect was dependent on the REST/NRSF transcriptional activity, as it was completely prevented by treatment with ODN (Fig. 8*D*). Immunoblotting analysis of GluA2 and VGLUT1 in total lysates of primary cortical neurons (Fig. 8*E*) corroborated the immunocytochemistry findings, demonstrating that 2DG treatment reduced GluA2 expression (Fig. 8*F*) without affecting VGLUT1 levels (Fig. 8*G*). Again REST/NRSF inhibition by ODN fully prevented GluA2 downregulation, confirming the involvement of REST/NRSF in this process (Fig. 8*E,F*).

To assess the functional consequences of reduced GluA2 expression we examined the contribution of GluA2-lacking AMPARs to evoked EPSCs in autaptic hippocampal neurons using IEM-1460, a selective blocker of GluA2-lacking AMPARs (100 μM). In control (vehicle/NEG-treated) neurons IEM-1460 reduced the EPSC amplitude by approximately 20% (Fig. 8*H,I*), consistent with the known low prevalence of GluA2-containing AMPARs in mature (14–16 div) excitatory neurons (Cull-Candy & Farrant, 2021; Liu & Zukin, 2007). However, in NEG/2DG-treated neurons the IEM-1460-induced block increased to approximately 50%, indicating a substantial increase in GluA2-lacking AMPARs (Fig. 8*H,I*). This electrophysiological evidence

independently validates the biochemical findings, supporting a REST/NRSF-dependent downregulation of the GluA2 subunit following glycolytic inhibition.

### Inhibition of glycolysis by 2DG suppresses spontaneous network activity, with REST/NRSF playing a significant but not exclusive role

To investigate the impact of glycolysis inhibition on the overall neuronal network activity, primary cortical neurons were treated with either NEG or ODN in the presence of vehicle (veh) or 2DG for 48 h (Fig. 9*A*). Spontaneous firing activity was then recorded using a 48 well multielectrode array (MEA; Fig. 9*A–C*). As previously reported (Prestigio et al., 2021), blocking REST/NRSF activity for 48 h with ODN had no effect on spontaneous network firing compared to NEG-treated neurons (Fig. 9*D–F*).

The effects of 2DG on network activity and its interaction with REST/NRSF were complex. Treatment with 2DG for 48 h significantly reduced mean firing rate, burst frequency, percentage of spikes within bursts and synchronization index (Fig. 9*D*). However, these inhibitory homeostatic effects were only partially REST/NRSF dependent, as they were only partially reversed by ODN treatment. In contrast, the 2DG-induced reductions in burst duration and network burst frequency were entirely REST/NRSF dependent, as they were fully blocked by ODN (Fig. 9*E*). Additionally, the effects of 2DG on network burst duration and burst inter-spike intervals were REST/NRSF independent, as they remained unchanged despite concomitant ODN treatment (Fig. 9*F*).

These findings demonstrate that glycolysis inhibition strongly suppresses neuronal network activity through a complex interplay of mechanisms, in which REST/NRSF plays a significant but not exclusive role.

### Discussion

Despite the wide availability of anti-seizure medications and their proven efficacy in controlling seizures, epilepsy cannot be considered an effectively curable neurological condition. Anti-seizure medications do not address the primary cause of seizures rooted in an E/I imbalance in the epileptic focus. Instead, they compensate for this imbalance by introducing an additional secondary functional alteration. Although effective in preventing seizures, this approach creates an artificial and often precarious equilibrium that significantly impacts the life quality of patients who benefit from seizure control but, in parallel, suffer from cognitive impairments, memory deficits, sedation and mood alterations, further exacerbating the psychological and social burden of

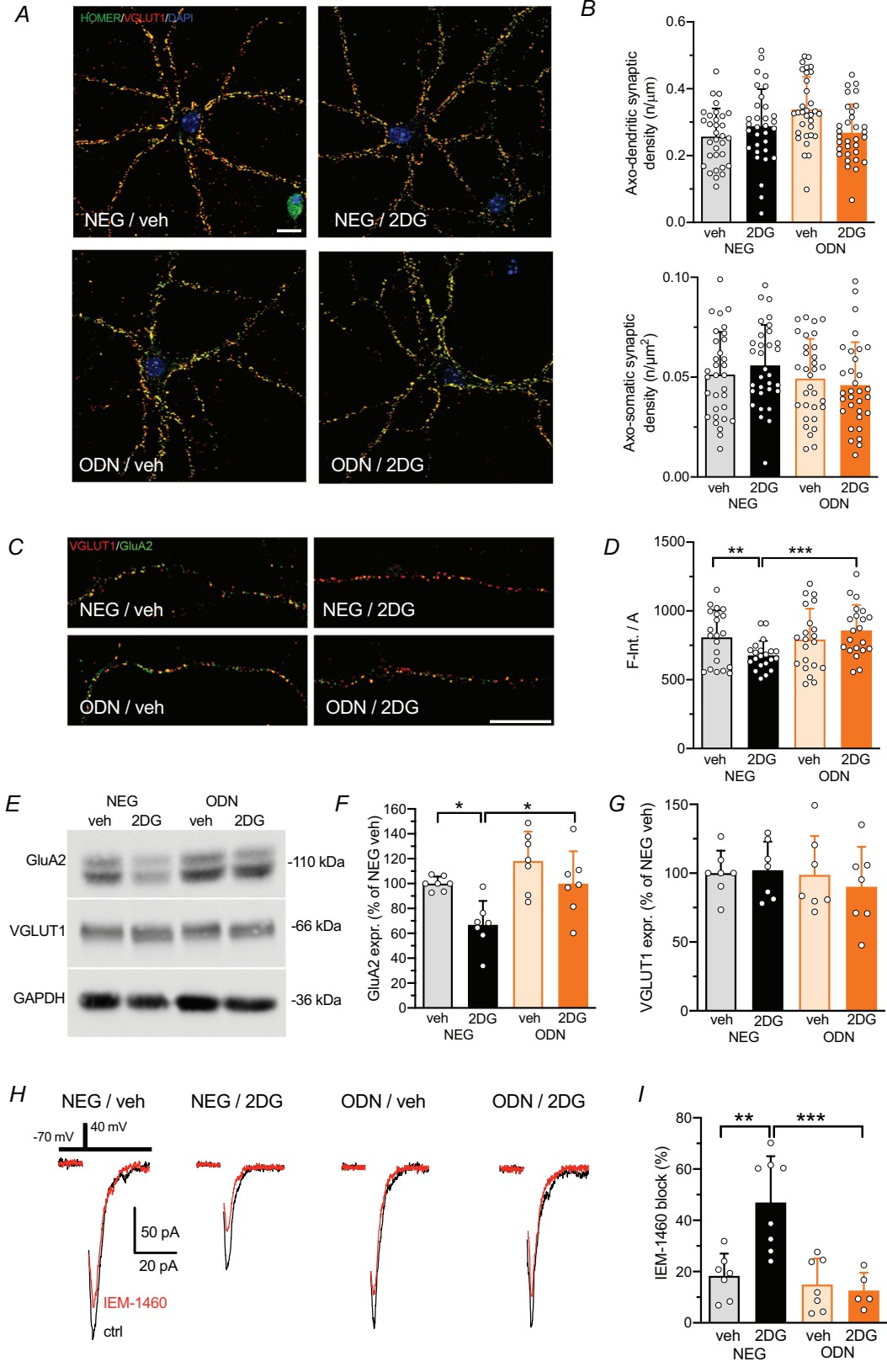

**Figure 8. 2DG treatment reduces the expression of GluA2-containing AMPA receptors at individual synaptic sites**

*A*, representative confocal images of hippocampal neurons (15 div) treated with NEG/veh, NEG/2DG, ODN/veh or ODN/2DG for 48 h. Neurons were labelled with Homer (*green*) and VGLUT1 (*red*) antibodies to identify *bona fide*

glutamatergic synapses. Scale bar, 20 μm. *B*, quantification of the density of axo-dendritic (*top*) and axo-somatic (*bottom*) synaptic contacts. Bar plots represent means ± SD with superimposed individual values. *n* = 30–32 fields from three independent neuronal preparations. Two-way ANOVA followed by the Tukey test. For axo-dendritic contacts: NEG/veh *vs.* NEG/2DG, *P* = 0.728. For axo-somatic contacts: NEG/veh *vs.* NEG/2DG, *P* = 0.944. *C*, representative confocal images of glutamatergic axo-dendritic synapses in primary hippocampal neurons (15 div) treated with NEG/veh, NEG/2DG, ODN/veh or ODN/2DG for 48 h, stained with VGLUT1 (*red*) and GluA2 (*green*) antibodies. Scale bar, 20 μm. *D*, quantification of the ratio between the integrated fluorescence intensity and the area of GluA2-positive puncta colocalizing with VGLUT1 immunoreactivity. Bar plots represent the means ± SD of *n* = 21 fields per condition from three independent neuronal preparations. Two-way ANOVA followed by Tukey's test. NEG/veh *vs.* NEG/2DG, *P* = 0.007; NEG/2DG *vs.* ODN/2DG, *P* = 0.0001. *E–G*, representative immunoblots (*E*) quantitative analysis of GluA2 (*F*) and VGLUT1 (*G*). Primary cortical neurons (15 div) were treated with NEG/veh, NEG/2DG, ODN/veh or ODN/2DG for 48 h. GAPDH was included as a loading control. Bar graphs show the mean ± SD (with superimposed individual values) immunoreactive signals, expressed in percentage of the NEG/veh control. *n* = 7 from three independent preparations. Two-way ANOVA followed by the Tukey test. For GluA2: NEG/veh *vs.* NEG/2DG, *P* = 0.025; NEG/2DG *vs.* ODN/2DG, *P* = 0.025. For VGLUT1: NEG/veh *vs.* NEG/2DG, *P* = 0.997; NEG/2DG *vs.* ODN/2DG, *P* = 0.721. *H*, representative traces of eEPSCs illustrating the inhibitory effect induced by perfusion of IEM-1460 (100 μM) in primary hippocampal neurons (14 div) pretreated for 48 h with NEG/vehicle, NEG/2DG, ODN/vehicle or ODN/2DG. *I*, quantification of the percentage inhibition induced by IEM-1460, presented as mean ± SD with individual superimposed data points. *n* = 5–8 cells from two independent cultures. Two-way ANOVA followed by the Tukey test. NEG/vehicle *vs.* NEG/2DG, *P* = 0.0011; NEG/2DG *vs.* ODN/2DG, *P* = 0.004.

epilepsy. Given these challenges, increasing attention has been directed towards dietary therapies, such as the KD and its variants MCT, MAD and LGID, that, by mimicking fasting, were shown to inhibit seizures. All the aforementioned dietary regimens are characterized by a severe reduction in carbohydrate intake, resulting in reduced or suppressed glycolysis. These considerations, together with the observation that minimal carbohydrate intake can abolish the seizure control achieved by KD (Huttenlocher, 1976), have led to a rapid increase in the studies investigating the anti-epileptic effects of agents that restrict glycolytic flux, primarily 2DG. This glucose analogue inhibits glycolysis, mimicking the anti-seizure effects of low-carbohydrate diets.

Numerous robust *in vivo* studies in murine and rat models have demonstrated the strong anti-seizure and disease-modifying effects of 2DG. Acute anti-convulsant effects have been reported in several seizure models, including 6 Hz electrically induced seizures ($ED_{50}$ ≈ 80 mg/kg in mice) (Sutula & Fountain, 2023), audiogenic seizures in Fring's mice ($ED_{50}$ ≈ 206 mg/kg) and chemically induced seizures, such as maximal electro-shock and PTZ models, in both rats and mice (Stafstrom et al., 2009). Moreover, chronic anti-epileptic effects of 2DG have been demonstrated in rat kindling models, where the long-term administration of 2DG attenuates seizure progression, elevates after-discharge thresholds and delays kindling development (Garriga-Canut et al., 2006). These effects are mediated by activation of REST/NRSF and recruitment of CtBP, leading to down-regulation of pro-epileptogenic genes such as *Bdnf* and *TrkB*. Importantly, in the traumatic brain injury model mice subjected to controlled cortical impact and treated acutely for 1 week with 2DG (250 mg/kg, intraperitoneally) exhibited reduced epileptiform activity, restoration of synaptic E/I balance and preservation of

parvalbumin-positive interneurons (Koenig et al., 2019). Regarding long-term safety chronic administration of 2DG (up to 6 months) in adult rats at doses up to 500 mg/kg/day demonstrated a favourable behavioural and systemic safety profile. No significant alterations were observed in the behavioural performance, cognitive functions or routine clinical parameters, indicating good tolerability of prolonged low-dose treatment (Ockuly et al., 2012). Furthermore clinical oncology studies have documented acceptable tolerability of 2DG in humans at doses ranging from 60 to 250 mg/kg, particularly under intermittent or low-dose regimens (Dwarakanath et al., 2009; Raez et al., 2013). These extensive preclinical findings are now being complemented by a phase II clinical trial (NCT05605301) currently investigating the safety and efficacy of 2DG in adults with drug-resistant epilepsy. Finally the combination of 2DG with metabolic therapies – such as KD, MCT or LGID – has been shown to enhance seizure control and further reduce glycolytic flux, thereby strengthening its translational potential (Rho & Boison, 2022; Rho et al., 2019). Although *in vivo* data clearly highlight the anti-seizure efficacy and tolerability of 2DG, its precise mechanism of action remains incompletely understood, and its variable effectiveness across experimental models requires further research to refine its therapeutic application.

In this study, we conducted a comprehensive analysis of the potential homeostatic effects activated by 2DG, specifically evaluating the contribution of the epigenetic modulator REST/NRSF among the mechanisms underlying the effects mediated by 2DG. We carried out this analysis under physiological conditions in neurons with normal levels of electrical activity, as it has been well established that the homeostatic action of REST/NRSF can be directly activated *in vitro* by conditions of neuronal hyperactivity (Pecoraro-Bisogni et al., 2018; Pozzi et al.,

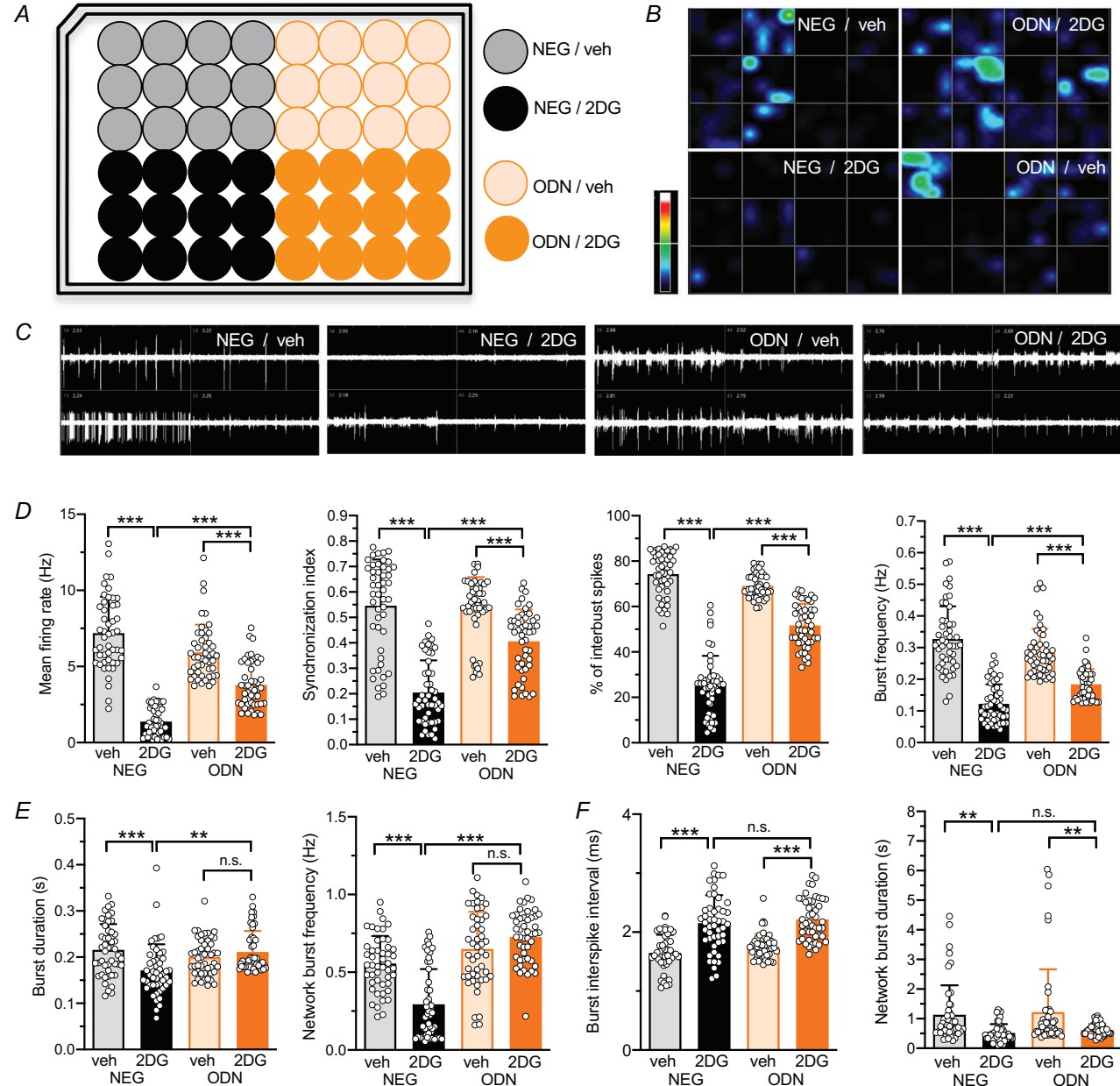

**Figure 9. 2DG treatment reduces the spontaneous activity of primary hippocampal networks in a partially REST/NRSF-dependent fashion**

*A*, representative multielectrode array (MEA) plate map showing the treatment group assignments: NEG/vehicle, NEG/2DG, ODN/vehicle and ODN/2DG. *B*, heat maps of firing rates of primary hippocampal networks (16–18 div) for each of the 48 wells in the multi-well MEA plate. *C*, representative 5 s traces of firing activity patterns from two electrodes across four representative wells, each corresponding to one of the four treatment groups after 48 h of treatment. *D*, bar plots representing the mean ± SD (with superimposed individual values) firing rate, synchronization index, percentage of spikes within bursts and burst frequency. *E*, bar plots represent the mean ± SD burst duration and network burst frequency. *F*, bar plots represent the mean ± SD burst inter-spike interval and network burst duration. $n = 48$ wells from four independent neuronal preparations. Two-way ANOVA followed by the Tukey test. The *P*-values of the indicated comparisons: NEG/vehicle *vs*. NEG/2DG, NEG/2DG *vs*. ODN/2DG and veh/ODN *vs*. ODN/2DG are reported below for each parameter: firing rate ($P < 0.0001$; $P < 0.0001$; $P < 0.0001$), synchronization index ($P < 0.001$; $P < 0.0001$; $P = 0.0002$), percentage of spikes within bursts ($P < 0.001$; $P < 0.0001$; $P < 0.0001$), burst frequency ($P < 0.001$; $P < 0.0001$; $P < 0.0001$), burst duration ($P = 0.002$; $P = 0.001$; $P = 0.704$), network burst frequency ($P < 0.001$; $P < 0.0001$; $P = 0.211$), burst inter-spike interval ($P < 0.001$; $P = 0.806$; $P < 0.0001$) and network burst duration ($P = 0.007$; $P = 0.935$; $P = 0.01$).

2013; Prestigio et al., 2021) and *in vivo* by various insults, such as seizures (Hu et al., 2011; Palm et al., 1998). Thus, the goal in this case was to evaluate the ability of 2DG to directly activate a homeostatic pathway mediated by REST/NRSF and to rule out the potential overlap with other homeostatic factors induced by neuronal hyperactivity, such as NPAS4 (Bloodgood et al., 2013).

Our data show that prolonged treatment with 2DG is well tolerated by hippocampal neurons *in vitro*, increases REST/NRSF expression and its translocation to the nucleus and induces a reduction in the NADH/NAD$^+$ ratio, as previously observed both in response to 2DG (Shen et al., 2017) and hypoglycaemia (Lee et al., 2024). It has been previously reported that reduced glycolysis, along with the resulting decrease in NADH levels, increases CtBP occupancy at the RE1 element of *Scn2A* and *BDNF*, effectively demonstrating that CtBP can enhance REST/NRSF-dependent repressor activity (Garriga-Canut et al., 2006). However, in our study we did not observe any downregulation of *BDNF* transcription, contrary to what was previously reported in the rat kindling model of temporal lobe epilepsy treated with 2DG (Garriga-Canut et al., 2006). This discrepancy is largely attributable to the substantial differences between the two experimental models. It is likely that the levels of REST/NRSF achieved in the kindling model, characterized by pronounced neuronal hyperactivity, are significantly higher compared to our experimental conditions. In that model REST/NRSF activation is driven both by the epileptogenic activity and by 2DG treatment, resulting in a combined effect capable of repressing a broader set of target genes, including *Bdnf*. By contrast in our study we used wild-type neurons, as our goal was to investigate the homeostatic processes triggered by 2DG in a healthy and intact neuronal network under basal conditions without any convulsant agents, where REST/NRSF levels are considerably lower. This may lead to a more moderate activation of REST/NRSF and, consequently, a more selective repression of its target genes. Indeed, it is well established that the repertoire of genes regulated by REST/NRSF depends on its activation levels, as different target genes exhibit variable affinities for REST/NRSF and its corepressors (McClelland et al., 2014).

A further unexpected result was the selectivity of the homeostatic effects of REST/NRSF induction by 2DG for the postsynaptic molecular machinery. Indeed, in previous works, we observed that under hyperactivity conditions induced by the convulsant agent 4-aminopyridine REST/NRSF activation reduces excitatory neuron firing frequency (Pozzi et al., 2013) and has opposite synaptic effects on excitatory (downscaling) and inhibitory (upscaling) synapses that are strictly implemented at the presynaptic level (Pecoraro-Bisogni et al., 2018; Prestigio et al., 2021). In this study, we did not observe any effect on the firing activity of excitatory

neurons nor did we detect changes in GABAergic inhibitory transmission. The only effect of REST/NRSF induction by 2DG was a downscaling of the strength of glutamatergic transmission at the postsynaptic level. The distinct homeostatic processes triggered by REST/NRSF activation under neuronal hyperactivity or glycolysis inhibition highlight how distinct stimuli can activate REST/NRSF in different ways, leading to different outcomes. This is likely due to different extents of REST/NRSF activation, as well as the recruitment of distinct REST/NRSF cofactors that result in differential transcriptional remodelling. These differences can justify why the final functional effects can substantially diverge.

Our data show that the postsynaptic mechanism of action, dependent on REST/NRSF activation, leads to downregulation of GluA2. This finding is supported by the absence of changes in mEPSC frequency and PPR and by cumulative amplitude analysis, all of which exclude alterations in neurotransmitter release probability. In parallel, western blot, immunocytochemistry and the increased magnitude of the IEM-1460-induced AMPAR blockade strongly implicate a downregulation of the GluA2 subunit of AMPAR as the primary mechanism underlying the reduced eEPSC amplitude. This effect mirrors previous observations in neuronal networks subjected to global ischaemia (Hwang & Zukin, 2018), where the activated REST/NRSF binds to the RE1 element in the GluA2 promoter, silencing the expression of the GluA2 subunit (Formisano et al., 2007; Noh et al., 2012). This is crucial because GluA2 serves as the '*ion gatekeeper*' of AMPARs, and AMPARs lacking GluA2 become permeable to $Ca^{2+}$ ions ($Ca^{2+}$ p-AMPARs). Notably, AMPARs lacking GluA2 can still function as signalling molecules and have been linked to various forms of synaptic plasticity, including homeostatic regulation (Ancona Esselmann et al., 2017; Yong et al., 2020). Thus, it is possible that 2DG activates two distinct REST/NRSF-dependent homeostatic pathways: (i) a decrease in the function of conventional AMPARs leading to reduced excitatory synaptic strength and (ii) an increased $Ca^{2+}$ influx through $Ca^{2+}$ p-AMPARs consequent to the downregulation of the GluA2 subunit, which could recruit additional homeostatic pathways.

It is also interesting to note that hippocampal principal neurons express high levels of GluA2, resulting in fast synaptic transmission with minimal $Ca^{2+}$ influx (Geiger et al., 1995). In contrast many GABAergic interneurons possess inwardly rectifying $Ca^{2+}$ p-AMPARs, which correlate with a low abundance of GluA2 mRNA (Jonas & Burnashev, 1995). Thus, the specific downregulation of the GluA2 subunit by REST/NRSF constitutes a homeostatic mechanism that counterbalances neural hyperexcitability by selectively reducing the synaptic strength of excitatory inputs onto principal neurons. Conversely this effect would be virtually absent at excitatory synapses

targeting inhibitory neurons due to their intrinsic lack of GluA2 subunits (Stincic and Frerking, 2015).

Despite the 2DG-induced downscaling of excitatory synaptic strength being entirely dependent on REST/NRSF, MEA recordings of spontaneous network activity reveal that the homeostatic effects of 2DG are not solely mediated by REST/NRSF induction. Specifically 2DG reduces burst duration and network burst frequency in a fully REST/NRSF-dependent manner. However, some key parameters, such as mean firing rate, synchronization index, percentage of intra-burst spikes and burst frequency, are only partially dependent on REST/NRSF activity, and other parameters, such as network burst duration and burst inter-spike interval, are totally REST/NRSF independent. This suggests that additional pathways contribute to the effects of 2DG on network dynamics, highlighting the complexity of the homeostatic mechanisms engaged by 2DG.

Previous studies have shown that 2DG inhibits the transcription of *Bdnf* and its receptor *TrkB*, both critical for kindling progression (Garriga-Canut et al., 2006); upregulates ATP-sensitive $K^+$ channels (Long et al., 2019; Lutas et al., 2014; Ma et al., 2007; Yang et al., 2013b), promoting membrane hyperpolarization and reducing excitability; enhances tonic GABAergic currents via neurosteroid synthesis (Forte et al., 2016); reduces oxidative stress (Yao et al., 2011); stimulates AMP-activated protein kinase (Potter et al., 2010). These multiple layers of action likely converge to suppress network hyperexcitability and abnormal synchrony, two hallmarks of epileptic activity. Taken together our findings suggest that the ability of 2DG to reduce network synchrony and excitability arises from both synaptic and intrinsic mechanisms, acting in parallel. This dual action may underlie its reported anti-seizure efficacy, as demonstrated in multiple preclinical studies, and provides a mechanistic framework for understanding how glycolysis inhibition dampens aberrant network activity.

Although an important feature of 2DG is its preferential uptake by metabolically active cells, targeting hyperexcitable brain regions in epilepsy (Sutula & Fountain, 2023), our study demonstrates that 2DG can also promote homeostatic effects in non-pathological neuronal networks. This is particularly interesting from a translational standpoint, as it suggests that the homeostatic action of 2DG could be preventive in situations that increase seizure propensity, such as trauma or infection. Thus although traditionally used in drug-resistant epilepsy, particularly in children, 2DG-based therapies should be reconsidered, in addition to epilepsy prevention, as a complementary approach to classic anti-seizure medications to enhance their effectiveness, reduce their dosage and associated side effects, ultimately improving patients' quality of life.

Although further studies are needed to fully define the 2DG therapeutic window, pharmacokinetics and long-term safety, early indications are encouraging. In particular an ongoing phase II clinical trial, and oncological therapies in which 2DG is tested in patients at clinically relevant doses, shows an overall favourable safety profile (Buono & Longo, 2018; Raez et al., 2013). The possible combination of 2DG with established metabolic interventions, such as the ketogenic or LGID, and the application of emerging drug delivery systems (e.g. brain-targeted nanocarriers) to increase brain specificity, adds further translational appeal. Taken together the available preclinical and early clinical data strongly support the concept of 2DG as a disease-modifying anti-seizure therapy. Although substantial work remains to be done, the converging lines of evidence are encouraging, making a compelling case for advancing this approach towards clinical development.

## Materials and methods

### Ethical approval

All animal procedures were performed in accordance with the guidelines of the European Community Council Directive 2010/63/EU and approved by the Italian Ministry of Health (authorization numbers: 253/2021-PR). All efforts were made to minimize animal suffering and to reduce the number of animals used. Pregnant C57BL/6J and Gad1-GFP heterozygous females were housed in individually ventilated cages in a temperature-controlled environment (22 ± 1°C), with a 12 h light/dark cycle and *ad libitum* access to food and water. No experimental procedures were conducted on live animals. Embryos (E17–E18) and postnatal mice (P0–P1) were killed by $CO_2$ inhalation followed by decapitation, as approved by the Institutional Animal Care and Use Committee. No anaesthesia was necessary for embryonic or early postnatal tissue collection.

### Blinding procedures

All experiments – including imaging, electrophysiological recordings, biochemical assays (western blot and qPCR) and multielectrode array (MEA) recordings, were performed under blinded conditions to prevent experimental bias. Treatment groups (2DG, ODN and NEG) were coded by an independent investigator not involved in data acquisition or analysis. Treatment codes were revealed only after all analyses were completed.

### Cell cultures

Primary cultures of hippocampal and cortical neurons were prepared from C57BL/6J embryos (Charles

River), as described previously (Valente et al., 2016a). Embryos at day 17–18 (E17/E18) were quickly removed by caesarean section, and hippocampi and cortices were dissected. Tissues were dissociated by enzymatic digestion in 0.125% trypsin for 20 min at 37°C and then triturated with a fire-polished Pasteur pipette and finally plated on Petri dishes that had been pre-treated with poly-L-lysine (0.1 mg/ml). All primary cultures were grown in a culture medium consisting of Neurobasal A (Gibco) supplemented with 2% B-27 (Invitrogen, Italy), 1 mM Glutamax, and 5 µg/ml Gentamycin and maintained at 37°C in a humidified incubator with 5% $CO_2$. Primary hippocampal neurons for experiments on glutamatergic/GABAergic autapses were prepared from postnatal *Gad1*-GFP knock-in mice (P0–P1), as previously described (Prestigio et al., 2019). *Gad1*-GFP knock-in mice were generated by inserting the cDNA encoding enhanced GFP into the Gad1 locus in TT2 embryonic stem cells, as described in Tamamaki et al. (2003). Heterozygous *Gad1*-GFP males were mated with wild-type C57BL6/J females, and GFP-positive pups were identified at birth through a Dual Fluorescent Protein Flashlight (DFP-1, NIGHTSEA, Lexington, MA) and confirmed by genotyping, performed by PCR with the following primers: TR-1b: GGCACAGCTCTC CCTTCTGTTTGC; TR-3: GCTCTCCTTTCGCGTTCCGACAG; TRGFP-8: CTGCTTGTCGGCCATGATAT AGACG. The autaptic neurons were prepared, as previously described, with a few minor modifications (Prestigio et al., 2019). Dissociated neurons were plated at very low density (20 cells/mm$^2$) on microdots (diameter of $\approx$ 40–300 mµ) obtained by spraying a mixture of poly-L-lysine (0.1 mg/ml dissolved in borate buffer 0.1 M; Sigma-Aldrich) and collagen (0.25 mg/ml in distilled water; Sigma-Aldrich, Milano, Italy) on Petri dishes (Falcon 35 mm, 353 001), pretreated the day before with 0.15% agarose. Under these culture conditions each Petri dish contained $\approx$ 15–20 isolated single autaptic neurons grown on poly-L-lysine micro-dots. The electrophysiological recordings of the synaptic transmission were carried out on single and isolated autaptic neurons between 10 and 15 div.

### Cell viability assay

Cell viability was assessed in low-density (160/mm$^2$) cultured hippocampal neurons (10–15 div) derived from C57BL/6J mice. The effect of glucose deprivation was tested over 24, 48, 72 and 96 h by reducing glucose concentrations from 25 to 10, 5 and 2 mM. To determine the optimal glucose/2DG ratio that does not compromise neuronal viability we assessed neuronal death by applying 2DG at a concentration twice that of glucose (2:1 ratio) at various glucose concentrations (10/20, 5/10, 2/4

and 1/2 mM). Finally we further evaluated neuronal viability in cultures maintained in 10 mM glucose and treated with 20 mM 2DG for 24, 48, 72 and 96 h. For both glucose deprivation and 2DG treatments cell viability was measured as follows: neurons were washed with prewarmed Tyrode solution, and cell nuclei were stained with 50 µg/ml Hoechst-33 342 (ThermoFisher R37605) for 20 min. Five minutes before the end of the Hoechst incubation, 200 ng/ml propidium iodide (PI) (Sigma-Aldrich P4170) was added. After the Hoechst and PI incubations neurons were washed again with Tyrode solution. Images were acquired using differential interference contrast and fluorescence microscopy with an Olympus IX71 microscope, equipped with a LCPlanFI 20× objective (Olympus, Tokyo, Japan), an ORCA-ER video camera (Hamamatsu Photonics, Hamamatsu City, Japan) and a Leica EL6000 fluorescence lamp (Leica Microsystems, Wetzlar, Germany). Image analysis was performed using ImageJ software. Cell mortality was quantified by counting Hoechst-positive nuclei (indicating all cells) and PI-positive nuclei (indicating dead cells), and the percentage of cell death was calculated as the ratio of PI-positive to Hoechst-positive puncta. The indicated sample size ($n$) corresponds to the number of fields per condition collected from three independent neuronal preparations.

### NAD$^+$/NADH determination

Cultured hippocampal neurons (14 div) were treated for 2 h with control medium supplemented with vehicle or with 10 mM 2DG. Cells ($10^5$ cells/well) were seeded in 12 well plates and lysed with 100 ml of 0.6 M perchloric acid or 0.1 M NaOH for NAD$^+$ or NADH determination, respectively. Cell extracts were centrifuged for 3 min at $16,000 \times g$; the supernatants were collected, and an aliquot was diluted 20-fold in 100 mM sodium phosphate buffer (pH 8.0) for the determination of NAD$^+$ content. The alkaline extracts were incubated at 70°C for 10 min and then diluted 20-fold in 10 mM Tris-HCl (pH 6). NAD$^+$ and NADH concentrations were estimated on neutralized extracts with a sensitive cycling enzymatic assay and normalized on protein content determined on cells seeded and treated in parallel (Sociali et al., 2019). Briefly the NAD(H) content in neutralized samples was assessed by adding 1 volume of a reaction mix to each sample, containing 2% (v/v) ethanol, 100 µg/ml alcohol dehydrogenase, 20 µM resazurin, 5 µg/ml diaphorase, 10 µM flavin mononucleotide, 10 mM nicotinamide, and 100 mM sodium phosphate (pH 8.0). The NAD reduced by alcohol dehydrogenases during the analysis is subsequently oxidized back to NAD by diaphorase, whereas the resazurin is reduced to resorufin, its fluorescent derivative. The accumulation of resorufin over time

was monitored every minute using a Fluostar Optima plate reader (BMG Labtechnologies GmbH, Offenburg, Germany).

## Oligodeoxynucleotides decoy and 2DG treatments

REST/NRSF activity was blocked using ODNs that act as surrogate binding sites for REST/NRSF and sequester the native transcription factor from its genomic binding sites (McClelland et al., 2011; Soldati et al., 2011). The ODN decoy (ODN) was designed to correspond to the canonical REST binding site, RE1, whereas a negative decoy control (NEG) was generated using a sequence corresponding to a non-canonical RE1 that does not bind REST/NRSF (Bruce et al., 2004). The decoy ODN sequences were as follows: ODN (positive decoy): (Top) 5′-GpPpCpPTPTTCAGCACCACGGACAGCGCCAGC-3′ (Bot) 3′ GpPpCpPTPGGCGCTGTCCGTGGTGCTGA AAGC-5′; NEG (negative decoy) (Top) 5′-GpPpCp PTPTCCAGCACAGTGGTCAGACCC-3′ (Bot) 3′-GpPp CpPTPTCTGACCACTGTGCTGGAAGC-5′. ODNs were designed with phosphorothioate modification on the first three nucleotides to avoid degradation (Lee et al., 2003; Osako et al., 2007). Single-stranded ODNs were synthesized by Sigma Genosys (St. Louis, MO, USA). Annealing was performed in $10\times$ buffer (100 mm Tris-HCl, pH 8.0, 10 mm EDTA and 1 m NaCl) by heating to at least 5–10°C above their melting temperature and cooling slowly using a heat block. Cultured neurons were treated with 200 nm decoys ODNs. For the live-cell imaging of activity-dependent REST/NRSF translocation from the cytoplasm to the nucleus, we used the following fluorescent ODNs: ODN-Top-Cy3: 5′-GpPpCpPTPTTCAGCACCACGGACAGCGCCAGC -Cy-3; NEG-Top-Cy3: 5′-GpPpCpPTPTCCAGCACAGT GGTCAGACCC-Cy3-3′. Hippocampal or cortical cultured neurons (12–15 div) were treated with: NEG/vehicle, NEG/2DG, ODN/vehicle or ODN/2DG. The treatment with 20 mm 2DG was performed on neurons maintained in 10 mm glucose. Where 2DG was not added (vehicle), 20 mm mannitol was included to maintain the extracellular osmolarity unchanged.

## Live imaging of the REST/NRSF translocation

Low-density (160 cells/mm$^2$) cultured hippocampal neurons (12–14 div) were treated with 10 mm glucose/20 mm 2DG for 4 h. After 2DG treatment, neurons were incubated for 24 h either with 200 nM ODN or NEG decoy tagged with cyanine-3 (Cy3-ODN; Cy3-NEG). Prior to image acquisition neurons were washed with prewarmed Tyrode solution, and cell nuclei were stained by incubating 50 μl/ml Hoechst-333 342 (R37605; ThermoFisher Scientific, Waltham, MA, USA).

The cells were further washed with Tyrode solution. Differential interference contrast and fluorescence images were acquired using an Olympus IX71 microscope with a LCPlanFI $40\times$ objective equipped with a Hamamatsu ORCA-ER camera and a Leica EL6000 fluorescence lamp. For each set of experiments exposure settings were maintained identical for all image acquisitions. For the Cy3 signal quantification analysis somatic and nuclear Cy3-positive regions of interest (ROI) were drawn for each neuron. The integrated fluorescence intensity (F-intensity) was measured as the sum of the fluorescence intensity values for each pixel of the cyto-solic or nuclear ROI. REST/NRSF partition between the cytosol and the nucleus was calculated from the ratio between $Cy3\_F\text{-intensity}_{nucleus}/Cy3\_F\text{-intensity}_{cytosol}$. Image analysis was performed using the ImageJ software. To be selected cells had to display at least 20% of the soma area covered by ODN-Cy3 fluorescence, ensuring unbiased selection based solely on effective probe uptake.

## Patch-clamp recordings

Electrophysiological experiments were performed at 13–15 div, after treatment with either 200 nm ODN or NEG in the presence of 20 mm 2DG or vehicle for 48 h. Whole-cell patch-clamp recordings were made at 10–20 kHz sampling rate and filtered at 1/5 of the acquisition rate with an eight-pole low-pass Bessel filter. Recordings with a leakage current > 100 pA or a series resistance > 15 MΩ were discarded. Data acquisition was performed using the PatchMaster software (HEKA Elektronik). Membrane potentials were not corrected for Donnan fluid junction potentials of 9 mV. All experiments were performed at room temperature (22–24°C). Patch pipettes made of thin borosilicate glass (Kimble, Kimax, Mexico) were drawn with a puller (P100, Narishige, Tokyo, Japan) and fire-polished with an MF2 microforge (Narishige Scientific Instruments) to a final resistance of 3–4 MΩ when filled with internal standard solution. The internal solution was as follows (in mM): 126 K gluconate, 4 NaCl, 1 MgSO$_4$, 0.02 CaCl$_2$, 0.1 BAPTA, 15 glucose, 5 HEPES, 3 ATP, 0.1 GTP, pH 7.2 (with KOH). The external solution contained (in mM) 140 NaCl, 2 CaCl$_2$, 1 MgCl$_2$, 4 KCl, 10 glucose, 10 HEPES, pH 7.3 (with NaOH).

For whole-cell recordings of eEPSCs in autaptic neurons (12–14 div), the external solution was supplemented with the following: D-(-)-2-amino-5-phosphonopentanoic acid (D-AP5, 50 μM; Tocris Bioscience, Bristol, UK) and bicuculline (30 μM; Tocris) to block NMDA and GABA$_A$ receptors, respectively. To record eIPSCs 6-Cyano-7-nitroquinoxaline-2,3-dione (CNQX, 10 μM; Tocris) and CGP 58 845 (10 μM; Tocris) were added to the external solution to block non-NMDA and GABA$_B$

receptors, respectively. Unclamped APs eliciting eE/IPSCs were activated by a brief depolarization of the cell body to 40 mV for 0.5 ms at 0.1 Hz. The eE/IPSCs were visually inspected, and only those events that were not contaminated by spontaneous activity were considered. To calculate the peak current during an isolated stimulus or a train of stimuli we first subtracted an averaged trace containing the stimulus artifact and the AP current without any detectable synaptic current (i.e. synaptic failures). For the analysis of the PPR two depolarizing pulses were applied with an inter-pulse interval of 50 ms. For each pair of E/IPSCs the PPR was calculated as the $I_2/I_1$ ratio, where $I_1$ and $I_2$ are the amplitudes of the eE/IPSCs evoked by the conditioning and test stimuli, respectively. In the analysis of synaptic responses during high-frequency stimulation (HFS) the inter-pulse interval was shorter than the time needed for an eEPSC to return to baseline, so eEPSCs overlapped partially. Therefore to correctly estimate the eEPSC amplitude the baseline of each event was defined as the final value of the decay phase of the preceding eEPSC, and the amplitude of eEPSC(n) was calculated by subtracting the residual amplitude of eEPSC(n-1) from its peak value. The size of the $RRP_{syn}$ and the probability that any given synaptic vesicle (SV) in the RRP will be released (Pr) were calculated using the cumulative amplitude analysis (Schneggenburger & Neher, 2000). The cumulative amplitude plot was determined by summing up peak EPSC amplitudes during 80 repetitive stimuli applied at 40 Hz. This analysis assumes that depression during the steady-state phase is limited by constant recycling of SVs, with an equilibrium occurring between released and recycled SVs, and that Pr during the train approaches the value 1. The cumulative amplitude profiles of the last 40 data points were fitted by linear regression and back-extrapolated to time 0. The intercept with the *y*-axis gave the $RRP_{syn}$ and the ratio between the amplitude of the first PSC ($I_1$) and the $RRP_{syn}$ yielded the Pr.

The $RRP_{tot}$ and the Pr were estimated by a second independent method: hyperosmotic stimulation with sucrose (Baldelli et al., 2007; Rosenmund & Stevens, 1996). Low-density hippocampal neurons (14–15 div) were voltage clamped at −70 mV, and maximal extracellular stimulation was delivered through current pulses of variable amplitude (40–80 μA; 0.2 ms duration) using an isolated pulse stimulator (Model 2100; AM-System Carlsburg, WA) to activate maximal eEPSC by recruiting all excitatory fibres projecting to the patched neuron. The same neuron was then challenged with a hypertonic solution (standard extracellular solution supplemented with 500 mм sucrose) delivered by a fast perfusion system. The extracellular solution was continuously perfused and could be rapidly switched (within 50–60 ms) to apply sucrose puffs for a controlled duration (6 s). The tip of the perfusion pipette (300 μm

diameter) was positioned approximately 150 μm from the soma. The charge transfer during the transient component of the synaptic current evoked by sucrose application was measured as the total readily releasable pool ($RRP_{tot}$) by integrating the current over a 2 s window. The release probability (Pr) was calculated as the ratio between the eEPSC charge – obtained by integrating the evoked current over 200 ms – and $RRP_{tot}$. As previously reported (Moulder & Mennerick, 2005), the $RRP_{tot}$ estimated by hyperosmotic sucrose application is typically five to six times larger than the $RRP_{syn}$ inferred from cumulative EPSC analysis during high-frequency stimulation. Consequently the Pr calculated from sucrose-based estimates appears lower. This discrepancy arises because sucrose triggers a massive, $Ca^{2+}$-independent release of all docked and fusion-competent vesicles, including those that are not readily recruited during physiological stimulation. In contrast cumulative analysis only captures the subset of vesicles that actively participate in $Ca^{2+}$-dependent and action potential-evoked release, thereby under-estimating the total size of the readily releasable pool.

mEPSCs were recorded in low-density (160 cells/mm$^2$) cultured hippocampal neurons (12–15 div) at a holding potential of −70 mV from low-density (160 cells/mm$^2$) hippocampal neurons in external solution supplemented with the following drugs: TTX (1 μм), D-AP5 (50 μм), CGP 58 845 (10 μм) and bicuculline (30 μм). Amplitude, frequency, half-width and rise time of mEPSCs were calculated using a peak detector function with a threshold amplitude set at 4 pA and a threshold area set at 50 ms*pA. mEPSC analysis was performed by using the Mini-analysis (Synaptosoft, Leonia, NJ, USA) and the Prism (GraphPad Software, Inc., Boston, MA, USA) software. Current-clamp recordings of neuronal firing activity were sampled at 50 kHz and low-pass filtered at 10 kHz in low-density (160 cells/mm$^2$) hippocampal neurons (12–15 div) using an external solution containing D-AP5 (50 μM), CNQX (10 μM), CGP (10 μM) and BIC (30 μM) to block NMDA, non-NMDA, $GABA_B$ and $GABA_A$ receptors, respectively. Resting membrane potential ($V_{rest}$) was determined after breaking the membrane and establishing the whole-cell mode. For the determination of evoked firing, neurons were current-clamped at −70 mV, and APs were elicited by 10 pA steps lasting 1000 ms applied every 2 s. The instantaneous firing frequency, calculated at the maximal value of injected current (250 pA), was estimated as the reciprocal value of the time difference between the first two evoked APs. The biophysical properties of the single AP were studied by measuring the time derivative of voltage (dV/dt) *vs*. voltage (phase-plane plot) of the first AP elicited by the minimal current injection. The phase-plan plot was used to extract the maximum rising slope, the AP peak,

the maximum repolarizing slope and the V threshold. The amplitude of APs was calculated as the difference between the $V_{max}$ and the threshold value. The half-width was calculated at half-maximal AP amplitude. The after-hyperpolarization (AHP) was defined as the most negative voltage reached immediately after the spike (Prestigio et al., 2019).

## Real-time qPCR

RNA was extracted using TRIzol reagent and purified on RNeasy spin columns (Qiagen NV, Venlo, The Netherlands). RNA samples were quantified at 260 nm with an ND1000 Nanodrop spectrophotometer (Thermo-Fisher Scientific). RNA purity was also determined by absorbance at 280 and 230 nm. All samples showed A260/280 and A260/230 ratios greater than 1.9. Reverse transcription was performed according to the manufacturer's recommendations on 1 μg of RNA with the QuantiTect Reverse Transcription Kit (Qiagen), which includes a genomic DNA-removal step. SYBR green RT-qPCR was performed in triplicate with 10 ng of template cDNA using QuantiTect Master Mix (Qiagen) on a 7900-HT Fast Real-Time System (Applied Biosystems), as previously described (Pozzi et al., 2013), using the following conditions: 5 min at 95°C, 40 cycles of denaturation at 95°C for 15 s and annealing/extension at 60°C for 30 s. Product specificity and occurrence of primer dimers were verified by melting-curve analysis. Primers were designed using Beacon Designer software (Premier Biosoft, Palo Alto, CA, USA) to avoid template secondary structure and significant cross-homology with other genes by BLAST search. The PCR reaction efficiency for each primer pair was calculated using the standard curve method with four serial dilution points of cDNA. The PCR efficiency calculated for each primer set was used for subsequent analysis. All experimental samples were detected within the linear range of the assay. Gene expression data were normalized using the multiple internal control gene method (Vandesompele et al., 2002) with the GeNorm algorithm available in qBasePlus software (Biogazelle, Zwijnaarde, Belgium). The housekeeping genes used were GAPDH and PPIA (peptidylprolyl isomerase); the expression of these genes was found not to be affected by the 2DG treatment. Primers sequences (5'–3') are indicated here:

*Rest_F* GAACCACCTCCCAGTATG
*Rest_R* CTTCTGACAATCCTCCATAG
*Bdnf cds_F* GATGCCGCAAACATGTCTATGA
*Bdnf cds_R* TAATACTGTCACACACGCTCAGCTC
Gapdh F GAACATCATCCCTGCATCCA
Gapdh_R CCAGTGAGCTTCCCGTTCA
Ppia_F CACTGTCGCTTTTCGCCGCTTG
Ppia_R TTTCTGCTGTCTTTGGAACTTTGTCTGC

## Protein extraction and western blotting

Total cell lysates were obtained from 14–15 div primary cortical neurons, plated at a density of 6000 cells/mm$^2$ and treated with either ODN or NEG (200 nм) in the presence of 2DG (20 mм) or vehicle for 24 or 48 h. After the treatment neurons were used to obtain total cell lysates. Lysis buffer (150 mм NaCl, 50 mм Tris-HCl pH 7.4, 1 mм EDTA, 1% Triton X100) containing a protease inhibitor cocktail (Cell Signalling Technology, Danvers, MA, USA) was used to prepare cell extracts. Cell lysates were collected and clarified by centrifugation for 10 min at 10,000 × $g$ at 4°C. Protein concentration was determined using the BCA assay (ThermoFisher). Equivalent protein amounts were subjected to SDS-PAGE on 8% polyacrylamide gels and blotted onto nitrocellulose membranes (Whatman, Maidstone, UK). Blotted membranes were blocked for 1 h using 5% milk in Tris-buffered saline (10 mм Tris, 150 mм NaCl, pH 8.0) together with 0.1% Triton X-100 and incubated 2 h at room temperature or overnight at 4°C with the following primary antibodies: rabbit anti-REST/NRSF (07-579; 1:500; Merck, Darmstadt, Germany), mouse anti-GluA2 (MAB397; 1:1,000; Merck), rabbit anti-VGLUT1 (135 303; 1:1,000; Synaptic System Goettingen, Germany), mouse anti-panNav (S8809; 1:300; Merck); rabbit anti-GAPDH (2118; 1:2,000; Cell Signalling Technology) antibodies. Membranes were washed and incubated for 1 h at room temperature with peroxidase-conjugated goat anti-mouse (706 516, 1:3,000; Bio-Rad, Hercules, CA) or anti-rabbit (1 706 515, 1:3,000; Bio-Rad) antibodies. Bands were revealed using the ECL chemiluminescence detection system (Bio-Rad), and the immunoreactivity quantification was performed by densitometric analysis of the fluorograms.

## Immunocytochemistry

Low-density (160 cells/mm$^2$) cultured hippocampal neurons (12–15 div) were treated with 20 mм 2DG or vehicle and either 200 nм ODN or NEG decoy for 48 h. Neurons were fixed for 10 min at room temperature with 4% paraformaldehyde/4% sucrose and then washed thrice with phosphate-buffered saline (PBS). After fixation neurons were permeabilized with 0.1% Triton X-100 for 10 min, and non-specific binding sites were blocked by 30 min incubation with a blocking solution composed of PBS supplemented with 5% fetal bovine serum (FBS). Incubation with primary antibodies in blocking solution was performed for 2 h at room temperature, followed by washing with PBS and final incubation with secondary antibodies for 1 h at room temperature. The following primary antibodies were used: mouse Homer (#160 011, 1:200, Synaptic Systems), guinea pig VGLUT1 (1:1,000, Synaptic Systems 135 304), rabbit GluA2

(#182 103, 1:500, Synaptic Systems). The secondary antibodies were Alexa488 goat anti-mouse (A11001, 1:500; Invitrogen/ThermoFisher Scientific), Alexa488 goat anti-rabbit (#A11034, 1:500, Invitrogen/ThermoFisher Scientific), Alexa568 goat anti-guinea pig (#A11075, 1:500; Invitrogen/ThermoFisher Scientific). After antibody staining samples were mounted in Prolong Gold Antifade reagent with 4′,6′-diamidino-2-phenylindole (DAPI) (#P36934; ThermoFisher Scientific). Confocal images were acquired using a TCS SP8 Confocal Laser Scanning Microscope with a $63\times$ oil-immersion objective (N.A. 1.4), equipped with hybrid detectors (Leica Microsystems). For each set of experiments exposure settings were maintained constant for all image acquisitions. Image analysis was performed using the ImageJ software.

For synapse identification the colocalization between presynaptic (VGLUT1) and postsynaptic (Homer or GluA2) markers was performed using the ImageJ plug-in JACoP by calculating the Manders' coefficient in fixed $3 \mu m^2$ ROIs (considered *bona fide* synapses) manually selected on dendrites and/or cell soma using the ImageJ selection tool. The synaptic density on dendrites and on the soma was obtained by counting the total number of synapses (positive VGLUT1/Homer colocalization puncta) per $\mu m$ of length for dendrites and per $\mu m^2$ for the soma. The indicated sample number ($n$) represents the number of fields collected from at least three independent neuronal preparations.

Postsynaptic GluA2-positive axo-dendritic glutamatergic synapses were identified as puncta showing colocalization of both VGLUT1 and GluA2. Postsynaptic GluA2 expression was estimated by calculating the ratio between the integrated fluorescence intensity (F-Int), measured as the sum of the fluorescence intensity values for each GluA2-positive pixel, and the area (A), expressed in $mm^2$, occupied by GluA2-positive signal (F-Int/A). At least 10 synapses per field were analysed. The indicated sample number ($n$) represents the number of images (fields) per condition collected from three independent neuronal preparations.

## MEA recordings

Neuronal activity was recorded using a multi-well MEA system (Maestro, Axion BioSystems, Atlanta, GA, USA). The MEA plates used (M768-tMEA-48W, Axion BioSystems) contain 48 wells, each with a square grid of 16 electrodes (50 $\mu m$ electrode diameter; 350 $\mu m$ centre-to-centre spacing) providing a recording area of $1.1\times1.1$ $mm^2$. MEAs, coated by depositing a 20 $\mu l$ drop of poly-L-lysine (0.1 mg/ml, Sigma-Aldrich) over each recording area, were incubated overnight. Dissociated hippocampal neurons were plated at a final density of 50,000 neurons per well in a 25 $\mu l$ drop covering an area of 22 $mm^2$, resulting in a final cell density of 2300

cells/$mm^2$. Neurons were incubated with neurobasal medium supplemented with 1% Glutamax, 2% B27 and 1% penicillin-streptomycin. One-third of the medium was replaced with fresh medium each week. Spiking activity from networks grown onto MEAs was recorded and monitored using Axion BioSystems hardware (Maestro1 amplifier and Middleman data acquisition interface) and Axion's Integrated Studio software in Spontaneous Neural Configuration (AxIS 2.4). Neuronal cultures on MEA plates were treated either with ODN or NEG (200 nM) in the presence of 2DG (20 mM) or vehicle at 12 div. After 48 h the MEA plates were placed on the Maestro apparatus, and their activity was recorded for 10 min at 37°C. After 1200 amplification raw data were digitized at 12.5 kHz/channel and stored for subsequent offline analysis. Spike detection and spike train data analysis were computed using the Axion BioSystems software NeuralMetricTool. To study the effect of treatments on firing and bursting properties only wells that contained $\geq4$ active electrodes (12 spikes/min) were considered for further analysis. Extracellular APs were detected by adaptive threshold crossing (5 times the SD of the rms noise on each channel) on 200 Hz high-pass-filtered traces. Bursts within single channels were identified by applying an inter-spike interval (ISI) threshold algorithm, which defines bursts as collections of a minimum number of spikes ($N_{min}$ = 5) separated by a maximum ISI ($ISI_{max}$) of 100 ms (Prestigio et al., 2021). Electrodes that recorded < 5 spikes/min were deemed inactive and were not included in the burst analysis. Network bursts were defined as a synchronized, rapid firing of neuronal activity across at least 4 of the 16 electrodes in the array.

## Statistical analysis

Data are given as means $\pm$ SD for $n$ = sample size. The normal distribution of experimental data was assessed using the Kolmogorov-Smirnov normality test. To compare two normally distributed sample groups the unpaired two-tailed Student's $t$ test was used. To compare two sample groups that were not normally distributed the Mann–Whitney $U$ test was used. To compare more than two normally distributed sample groups we used one- or two-way ANOVA followed by Dunnett's or Tukey's multiple comparison tests, respectively. Alpha levels for all tests were 0.5% (95% confidence intervals). To compare more than two not normally distributed sample groups we used Kruskal-Wallis followed by Dunn's multiple comparison test. In the figures statistical significance is indicated as follows: $*P < 0.05$, $**P < 0.001$, $***P < 0.0001$ and $ns$ when $P > 0.05$. Statistical analysis was carried out using SPSS (version 21, IBM Software Armonk, NY, USA), Prism software (GraphPad Software, Inc., Boston, MA, USA) and OriginPro-8 (OriginLab Corp., Northampton, MA, USA).

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

## Additional information

### Data availability statement

The datasets generated and analysed during the current study are available from the corresponding author on reasonable request.

### Competing interests

The authors have no relevant financial or non-financial interests to disclose.

### Author contributions

D.F. performed patch-clamp and MEA recordings, analysed data and prepared figures; B.P. conducted optical and confocal imaging experiments, analysed data and prepared figures; A.B. and S.B. performed NAD+/NADH experiments, analysed data and prepared figures; F.O. and A.M. performed immunoblot experiments, analysed data and prepared figures; C.P. conducted PCR experiments, analysed data and prepared figures; M.D. contributed to electrophysiological experiments, data analysis and preparation of figures; F.B., F.O., P.V. and P.B. discussed the data and funded the research project. P.V. supervised the research, analysed data and prepared figures. P.B. designed and supervised the research, analysed data, prepared figures and wrote the manuscript. All authors reviewed and revised the manuscript.

### Funding

Open access funding was provided by Università degli Studi di Genova within the CRUI - CARE Agreement. The study was supported by research grants from the Italian Ministry of University and Research: NEXTGENERATIONEU (NGEU) and funded by the Ministry of University and Research (MUR), National Recovery and Resilience Plan (NRRP), project MNESYS (PE0000006) - A multiscale integrated approach to the study of nervous system in health and disease - (DN. 1553 11.10.2022 to F.O. and P.B.; PRIN2020XBFEMS to P.B.; PRIN-PNRR-P2022CRAXJ to P.B.; PRIN2022MPCKWW to P.V. The study was also supported by research grants from IRCCS Ospedale Policlinico San Martino Genova Ricerca Corrente and 5×1000 grants to P.B. and from the Italian Ministry of Health: Ricerca Finalizzata 2021–12 372 526 to P.B.

### Acknowledgements

The authors would like to thank Dr Silvia Casagrande (Department of Experimental Medicine, University of Genova, Genova, Italy) for the preparation of primary hippocampal neuronal cultures.

Open access publishing facilitated by Universita degli Studi di Genova, as part of the Wiley - CRUI-CARE agreement.

### Keywords

epigenetic modulator, epilepsy, excitatory synapses, glycolysis, homeostatic plasticity, postsynaptic mechanism

### Supporting information

Additional supporting information can be found online in the Supporting Information section at the end of the HTML view of the article. Supporting information files available:

**Peer Review History**

