## [Peer Review History · The Journal of Physiology]

Harnessing Metabolic Control for Synaptic Stability: REST/NRSF Links Glycolytic Inhibition to Excitatory Neurotransmission

Pietro Baldelli, Pierluigi Valente, Franco Onofri, Santina Bruzzone, Fabio Benfenati, Daniele Ferrante, Barbara Parisi, Antonella Marte, Dionisi Marianna, Andrea Benzi, and Cosimo Prestigio

DOI: 10.1113/JP288763

Corresponding author(s): Pietro Baldelli (pietro.baldelli@unige.it)

The following individual(s) involved in review of this submission have agreed to reveal their identity: Jaideep Kapur (Referee #2)

Review Timeline:

Submission Date:	19-Feb-2025
Editorial Decision:	31-Mar-2025
Revision Received:	23-Jul-2025
Accepted:	05-Sep-2025

Senior Editor: Katalin Toth

Reviewing Editor: Gareth Morris

Transaction Report:

Dear Dr Baldelli,

Re: JP-RP-2025-288763 "Harnessing Metabolic Control for Synaptic Stability: REST/NRSF Links Glycolytic Inhibition to Excitatory Neurotransmission" by Pietro Baldelli, Pierluigi Valente, Franco Onofri, Santina Bruzzone, Fabio Benfenati, Daniele Ferrante, Barbara Parisi, Antonella Marte, Dionisi Marianna, Andrea Benzi, and Cosimo Prestigio

Thank you for submitting your manuscript to The Journal of Physiology. It has been assessed by a Reviewing Editor and by 2 expert referees and we are pleased to tell you that it is potentially acceptable for publication following satisfactory major revision.

REVISION CHECKLIST:

We look forward to receiving your revised submission.

Yours sincerely,

Katalin Toth
Senior Editor
The Journal of Physiology

REQUIRED ITEMS

- Include a Key Points list in the article itself, before the Abstract.
- You must start the Methods section with a paragraph headed Ethical approval (https://jp.msubmit.net/cgi-bin/main.plex?form_type=display_requirements#methods).

Research must comply with The Journal's policies regarding animal experiments (<https://physoc.onlinelibrary.wiley.com/hub/animal-experiments>) and adherence to these policies must be stated in the manuscript.

Authors should confirm in their Methods section that their experiments were carried out according to the guidelines laid down by their institution's animal welfare committee, including an ethics approval reference number. The Methods section must contain a statement about access to food, water and housing, details of the anaesthetic regime: anaesthetic used, dose and route of administration, and method of killing the experimental animals.

- The reference list must be in alphabetical order, rather than numbered, to comply with our Journal format.
- Please ensure that any tables are editable and in Word format, and wherever possible, embedded in the article file itself.
- Please ensure that the Article File you upload is a Word file.
- Papers must comply with the Statistics Policy: https://jp.msubmit.net/cgi-bin/main.plex?form_type=display_requirements#statistics.

In summary:

- If n {less than or equal to} 30, all data points must be plotted in the figure in a way that reveals their range and distribution. A bar graph with data points overlaid, a box and whisker plot or a violin plot (preferably with data points included) are acceptable formats.
- If $n > 30$, then the entire raw dataset must be made available either as supporting information, or hosted on a not-for-profit repository, e.g. FigShare, with access details provided in the manuscript.
- 'n' clearly defined (e.g. x cells from y slices in z animals) in the Methods. Authors should be mindful of pseudoreplication.

- All relevant 'n' values must be clearly stated in the main text, figures and tables.
- The most appropriate summary statistic (e.g. mean or median and standard deviation) must be used. Standard Error of the Mean (SEM) alone is not permitted.
- Exact p values must be stated. Authors must not use 'greater than' or 'less than'. Exact p values must be stated to three significant figures even when 'no statistical significance' is claimed.

EDITOR COMMENTS

Reviewing Editor:

Methods Details:

Some important method details are missing, as Reviewer #1 suggested.

Comments to the Author:

This manuscript explores the critical role of REST/NRSF in the effects of glycolysis inhibition on neuronal activity, and provides a potential strategy for treating epilepsy. Both reviewers acknowledge the study's innovation and scientific value but highlight several areas for improvement. Please improve the manuscript according to these comments, including in vivo validation, the effects of 2-DG on inhibitory synaptic transmission, potential toxicity and dose equivalence concerns related to 2-DG, GluA2 expression and presynaptic effects, and providing more detailed methods.

REFEREE COMMENTS

Referee #1:

The study is well-designed, rigorous in methodology, and innovative in its findings, providing important insights into the role of metabolic regulation in neuronal activity. The results hold potential clinical significance for developing new anti-epileptic therapies. It is recommended that the authors revise the manuscript according to the above suggestions before acceptance for publication. The revised manuscript will be more convincing and provide an important reference for research in related fields. Please find detailed comments in the attached document.

Referee #2:

Inhibiting glycolysis with 2-deoxyglucose (2-DG) suppresses seizures acutely and on chronic treatment. Mechanism of anti-seizure action remains under investigation. The author studied the action of 20 mM 2-DG applied to cultured neurons for up to 24 hours. They studied the role of REST/NRSF in the effects of glycolysis inhibition on hippocampal neuron activity. Treatment with 2-deoxy-D-glucose (2DG) decreased the NADH/NAD⁺ ratio, increased REST/NRSF expression, and promoted its nuclear translocation. The amplitude of evoked (eEPSCs) and miniature (mEPSCs) excitatory postsynaptic currents was reduced in a REST/NRSF-dependent manner. This effect was associated with a REST/NRSF-dependent reduction in the size of GluA2-positive puncta and a decrease in GluA2 expression without changes in the density of excitatory synapses.

1) Recent studies in experimental animals have revealed that 2-DG has cardiac toxicity doses above 60 mg/Kg. See (Terse PS, Joshi PS, Bordelon NR, Brys AM, Patton KM, Arndt TP, Sutula TP. 2-Deoxy-d-Glucose (2-DG)-Induced Cardiac Toxicity in Rat: NT-proBNP and BNP as Potential Early Cardiac Safety Biomarkers. *Int J Toxicol.* 2016 May;35(3):284-93. doi: 10.1177/1091581815624397. Epub 2016 Feb 2. PMID: 26838190; PMCID: PMC4864115.). A dose of 37.5 mg/Kg is safe. The authors need to address this issue. What is the dose equivalent of 20 mM 2 DG?

2) These studies are performed in cultured neurons with many autaptic connections. Therefore, these results may not apply in vivo.

3) The authors argue that 2 DG has non-presynaptic effects on synaptic transmission. They interpreted their data to suggest that release probability was unchanged. However, this conclusion should be confirmed by measuring release probability by another method, e.g., the method of failures.

4) The authors claim that EPSC amplitude is reduced because of reduced GluA2 expression. GluA2 lacking receptors rectify, and are sensitive to IEM and Philanthotoxin. Authors need to confirm their findings of diminished GluA2 expression with electrophysiology.

Minor:

- 1) Please specify if the experimenter was blinded to 2 DG/ODN treatment.
- 2) How was sampling bias addressed when selecting cells to study REST relocation to the nucleus?
- 3) Drugs used to treat epileptic seizures were called Anti epileptic drugs but are best referred to as Antiseizure medications.

END OF COMMENTS

Review Comments

Manuscript Title: Harnessing Metabolic Control for Synaptic Stability: REST/NRSF Links Glycolytic Inhibition to Excitatory Neurotransmission

Manuscript Number: JP-RP-2025-288763

This study explores the regulatory mechanisms of metabolic control on synaptic stability and thoroughly investigates the intrinsic link between REST/NRSF, glycolytic inhibition, and excitatory neurotransmission. It successfully reveals key molecular mechanisms and provides new directions and potential therapeutic targets for the study of epilepsy and other neurological diseases. The findings are of great significance in the field of epilepsy. Overall, the manuscript is innovative and scientifically valuable and meets the publication requirements of The Journal of Physiology. However, to further enhance the quality of the research and its academic impact, the authors are advised to make detailed revisions to the following aspects:

1. Clinical Translation Potential

The manuscript highlights the potential therapeutic effects of 2DG on epilepsy but lacks sufficient discussion on its prospects and challenges in clinical applications. The study is mainly based on in vitro experiments and lacks in vivo data (e.g., from epilepsy animal models) to validate the anti-epileptic effects of 2DG. Given that the study is largely based on in vitro experiments, it would be beneficial to include in vivo data from epilepsy animal models to substantiate the anti-epileptic effects of 2DG.

Additionally, the study uses a 2DG concentration of 20 mM but does not investigate the effects of different concentrations. Dose-dependent studies would help better understand the mechanisms of 2DG. Conducting dose-dependent studies could provide deeper insights into the mechanisms underlying 2DG's efficacy.

2. Excitatory and Inhibitory Balance

Although the study provides valuable insights into the downregulation of excitatory synaptic transmission, it overlooks the potential impact on inhibitory synapses. It is recommended to supplement the analysis of inhibitory synaptic transmission to explore whether 2DG achieves its

anti-epileptic effects by modulating the balance between excitatory and inhibitory transmission. The MEA results show that 2DG significantly reduces network activity, but the underlying mechanisms are not thoroughly discussed. It is suggested to analyze in the Discussion how 2DG regulates network synchrony by altering synaptic strength and neuronal excitability and whether this regulation is directly related to the inhibition of epileptic seizures.

3. Multilevel Validation

While the study examines the effects of 2DG at the molecular, cellular, and network levels, it lacks assessments of behavioral or cognitive functions. It is recommended to supplement behavioral experiments (e.g., seizure frequency, cognitive function tests) to validate the therapeutic efficacy of 2DG in whole-animal models.

4. Figures and Legends

The legends are not detailed enough, and some do not clearly explain experimental conditions, statistical methods, or significance markers. Ensuring consistency between figure numbering and citations would improve the manuscript's clarity. The manuscript also contains inconsistencies in figure numbering and citations (e.g., the order of Figure 4 and Figure 5 is confused). It is recommended to check and correct the consistency between figure numbering and citations.

5. Methods Section

Some experimental details are missing, and certain methods are not described in sufficient detail. Providing the detailed descriptions of the experimental methods to ensure reproducibility. For example:

Clearly describe the environmental parameters for cell culture (e.g., temperature, CO₂ concentration) and the specific components of the culture medium to ensure reproducibility.

In electrophysiological experiments, explicitly describe the preparation methods of electrodes, specific recording parameters (e.g., sampling rate, filtering range), and stimulation protocols used in the experiments.

Provide the sources of antibodies, dilution factors, and specific experimental steps (e.g.,

incubation times and temperatures).

Overall, the study is well-designed, rigorous in methodology, and innovative in its findings, providing important insights into the role of metabolic regulation in neuronal excitability. The results hold potential clinical significance for developing new anti-epileptic therapies. It is recommended that the authors revise the manuscript according to the above suggestions before acceptance for publication. The revised manuscript will be more convincing and provide an important reference for research in related fields.

Point-by-Point Response to Reviewers

REFEREE #1:

The study is well-designed, rigorous in methodology, and innovative in its findings, providing important insights into the role of metabolic regulation in neuronal activity. The results hold potential clinical significance for developing new anti-epileptic therapies. It is recommended that the authors revise the manuscript according to the above suggestions before acceptance for publication. The revised manuscript will be more convincing and provide an important reference for research in related fields. Please find detailed comments in the attached document.

This study explores the regulatory mechanisms of metabolic control on synaptic stability and thoroughly investigates the intrinsic link between REST/NRSF, glycolytic inhibition, and excitatory neurotransmission. It successfully reveals key molecular mechanisms and provides new directions and potential therapeutic targets for the study of epilepsy and other neurological diseases. The findings are of great significance in the field of epilepsy. Overall, the manuscript is innovative and scientifically valuable and meets the publication requirements of The Journal of Physiology.

However, to further enhance the quality of the research and its academic impact, the authors are advised to make detailed revisions to the following aspects:

- We thank the Reviewer for the thoughtful and constructive criticisms and positive comments.

1. Clinical Translation Potential The manuscript highlights the potential therapeutic effects of 2DG on epilepsy but lacks sufficient discussion on its prospects and challenges in clinical applications.

1- In the revised manuscript, we have substantially expanded the Discussion section to provide a more comprehensive and forward-looking analysis of the translational potential of 2DG in epilepsy. Specifically, we now highlight both the robust preclinical evidence and the emerging clinical perspectives, as summarized below:

- We discuss the well-established antiepileptic and neuroprotective effects of 2DG in multiple animal models of both acquired and genetic epilepsies, emphasizing consistent and reproducible findings from independent studies that support its potential as a disease-modifying treatment for pharmaco-resistant epilepsy.
- We note that 2DG crosses the blood–brain barrier via GLUT1 transporters and accumulates in neurons and glial cells, making it a pharmacologically accessible target for central nervous system interventions.
- We acknowledge key challenges for clinical translation, including the need for better characterization of the therapeutic window and long-term safety in humans, the optimization of dosing regimens to minimize systemic toxicity, and the lack of current large-scale clinical trials in epilepsy.
- We note that a Phase II clinical trial is currently underway to evaluate the safety and tolerability of 2DG in adults with drug-resistant epilepsy. We also reference prior oncology trials, which demonstrated a generally favorable safety profile for 2DG, particularly under intermittent or low-dose regimens.
- We discuss the potential combination of 2DG with metabolic therapies, such as ketogenic or low-glycemic index diets, to enhance therapeutic efficacy while reducing systemic exposure. We also mention ongoing efforts to improve brain selectivity through advanced drug delivery strategies.

All these aspects are now explicitly addressed in the revised Discussion section and are supported by an extensive list of references.

2. The study is mainly based on in vitro experiments and lacks in vivo data (e.g., from epilepsy animal models) to validate the anti-epileptic effects of 2DG. Given that the study is largely

based on *in vitro* experiments, it would be beneficial to include *in vivo* data from epilepsy animal models to substantiate the anti-epileptic effects of 2DG.

2- We fully understand the Reviewer's concern regarding the absence of *in vivo* data. We respectfully acknowledge that the inclusion of *in vivo* experiments, such as behavioral or electroencephalographic seizure assessments, would undoubtedly provide valuable translational insight. However, as noted in our communication with the Senior and Reviewing Editors, in the Italian regulatory context, the authorization process for animal experiments involving epileptic models typically requires 10 to 12 months, making it incompatible with the current revision timeline. For this reason, we are unable to include new *in vivo* experiments at this stage.

Moreover, we would like to point out that the antiepileptic efficacy of 2DG has been extensively validated in numerous animal models of epilepsy, as documented by multiple independent studies across a wide range of paradigms (Garriga-Canut *et al.*, 2006; Stafstrom *et al.*, 2009; Gasior *et al.*, 2010; Ockuly *et al.*, 2012; Yang *et al.*, 2013; Bazzigaluppi *et al.*, 2017; Shao *et al.*, 2018; Koenig *et al.*, 2019; Nikbakht *et al.*, 2019; Janicot *et al.*, 2020; Sutula & Fountain, 2023). These include models of acquired epilepsy (e.g., kainate, pilocarpine, kindling), as well as models of genetic epileptic encephalopathies. A comprehensive list of references supporting these findings has been added to the Discussion (see new citations), and the relevant section has been substantially revised and expanded to reflect the current *in vivo* knowledge.

We now highlight that 2DG:

- exhibits robust anticonvulsant effects in several established seizure models;
- exerts disease-modifying actions when administered after seizure onset;
- has entered a phase II clinical trial (NCT05605301) aimed at evaluating its safety, tolerability, and pharmacokinetics in patients with pharmaco-resistant epilepsy.

We also discuss how our work contributes mechanistically to this body of evidence by identifying REST/NRSF as a downstream effector of 2DG in neuronal cultures, offering a molecular framework that may explain some of its antiepileptic properties observed *in vivo*. On the other hand, we agree with the Reviewer that mechanistic links between 2DG, REST/NRSF, and behavioral outcomes *in vivo* remain underexplored, and we now explicitly state this in the revised manuscript as an important direction for future work.

3. Additionally, the study uses a 2DG concentration of 20 mM but does not investigate the effects of different concentrations. Dose-dependent studies would help better understand the mechanisms of 2DG. Conducting dose-dependent studies could provide deeper insights into the mechanisms underlying 2DG's efficacy.

3- We fully agree that dose–response studies are essential for clarifying concentration-dependent mechanisms of action. In our study, the selection of 10 mM glucose and 20 mM 2DG was based on both methodological and biological considerations.

Firstly, it is important to emphasize that primary neurons *in vitro* require relatively high glucose concentrations (typically 25 mM) to ensure survival, far above the physiological cerebrospinal fluid levels (1–2 mM). This necessity arises from several factors: (i) the high vulnerability of cultured neurons to suboptimal environmental conditions; (ii) enhanced oxidative stress *in vitro* compared to *in vivo*; (iii) elevated metabolic demand due to continuous synaptic activity and ongoing network formation; (iv) absence of alternative substrates such as lactate or ketone bodies; and (v) lack of glial metabolic support.

Secondly, to achieve an effective competition between glucose and 2DG under our reduced glucose condition (10 mM glucose), we used a 2:1 ratio (20 mM 2DG), in line with prior literature and theoretical models of competitive inhibition at the levels of hexokinase and glucose transport (Garriga-Canut *et al.*, 2006; Zhao *et al.*, 2019; Pajak *et al.*, 2020; Harders *et al.*, 2024). This approach ensures robust and reproducible glycolytic inhibition.

To directly address the Reviewer's request, we performed dose–response viability experiments using several glucose/2DG ratios (10/20, 5/10, 2/4, and 1/2 mM). These results, now presented in the new panel B of Figure 1, demonstrate that neuronal viability was optimal at 10/20 mM and that the 1:2 glucose/2DG ratios at lower glucose concentrations led to progressively increased neuronal death. Thus, our experimental conditions represent the least toxic combination capable of effectively inhibiting glycolysis, thereby minimizing confounding effects from nonspecific stress or cytotoxicity.

This point is especially critical for our study, which focuses on REST-mediated regulatory mechanisms. As REST can be activated by cellular stress, including excitotoxicity (Palm et al., 1998; Noh et al., 2012; Kaneko et al. 2014; Buffolo et al., 2021), our goal was to study REST activation while preserving neuronal viability and network function.

4. **Excitatory and Inhibitory Balance** Although the study provides valuable insights into the downregulation of excitatory synaptic transmission, it overlooks the potential impact on inhibitory synapses. It is recommended to supplement the analysis of inhibitory synaptic transmission to explore whether 2DG achieves its anti-epileptic effects by modulating the balance between excitatory and inhibitory transmission.

4- This point was already addressed in Figure 5 of the former version of the manuscript. Specifically, we performed patch-clamp recordings from autaptic hippocampal neurons cultured from GAD67-GFP mice to distinguish between excitatory and inhibitory neurons. Our data (Figure 5A–D) show that 48-hour treatment with 2DG selectively reduces the amplitude of evoked excitatory postsynaptic currents (eEPSCs) in GFP-negative excitatory neurons, with no detectable effect on the amplitude or paired-pulse ratio (PPR) of evoked inhibitory postsynaptic currents (eIPSCs) recorded from GFP-positive inhibitory neurons. These results indicate that glycolysis inhibition by 2DG downscaling excitatory synaptic strength via a REST/NRSF-dependent mechanism, without affecting inhibitory synaptic transmission. We better emphasize this point in the Results and Discussion sections.

5. **The MEA results** show that 2DG significantly reduces network activity, but the underlying mechanisms are not thoroughly discussed. It is suggested to analyze in the Discussion how 2DG regulates network synchrony by altering synaptic strength and neuronal excitability and whether this regulation is directly related to the inhibition of epileptic seizures.

5-We thank the Reviewer for this important comment. We have now substantially expanded the Discussion to better address the mechanisms by which 2DG regulates network synchrony and excitability. Although the REST/NRSF-dependent downscaling of excitatory synaptic strength fully accounts for some network parameters, the MEA data reveal additional REST/NRSF-independent mechanisms. We now discuss how 2DG modulates both synaptic and intrinsic neuronal properties, likely through a combination of transcriptional and metabolic effects, including modulation of ion channels, neurosteroidogenesis, oxidative stress, and energy metabolism, as supported by previous literature. We also explicitly discuss how these combined mechanisms may converge to suppress pathological network synchronization, potentially contributing to the anti-epileptic actions of 2DG.

6. **Multilevel Validation** While the study examines the effects of 2DG at the molecular, cellular, and network levels, it lacks assessments of behavioral or cognitive functions. It is recommended to supplement behavioral experiments (e.g., seizure frequency, cognitive function tests) to validate the therapeutic efficacy of 2DG in whole-animal models.

6- We appreciate the Reviewer's recommendation. However, we discussed and agreed with the Senior and Reviewing Editors that performing new in vivo experiments is not feasible within the current revision timeline (see response to Point 2). As mentioned above, the anti-epileptic and disease-modifying efficacy of 2DG has already been thoroughly validated in a wide range of animal models of epilepsy by numerous independent studies. We agree with the Reviewer that future studies exploring the behavioral and cognitive consequences of 2DG-mediated mechanisms in vivo would be highly valuable, and we now explicitly highlight this as an important area for future research in the revised Discussion.

7. **Figures and Legends.** The legends are not detailed enough, and some do not clearly explain experimental conditions, statistical methods, or significance markers. Ensuring consistency between figure numbering and citations would improve the manuscript's clarity. The manuscript also contains inconsistencies in figure numbering and citations (e.g., the order of Figure 4 and Figure 5 is confused). It is recommended to check and correct the consistency between figure numbering and citations.

7- We carefully reviewed the entire manuscript to ensure full consistency between figure numbering and in-text citations, with particular attention to Figures 4 and 5. Furthermore, we revised and improved all figure legends by integrating additional details regarding experimental conditions, including treatment concentrations and durations, as well as relevant genotypes and experimental models, where applicable. We also clarified the statistical methods applied in each experiment and explicitly indicated the meaning of significance markers (e.g., *p* values and tests used).

8. Methods Section Some experimental details are missing, and certain methods are not described in sufficient detail. Providing the detailed descriptions of the experimental methods to ensure reproducibility. For example: Clearly describe the environmental parameters for cell culture (e.g., temperature, CO₂ concentration) and the specific components of the culture medium to ensure reproducibility. In electrophysiological experiments, explicitly describe the preparation methods of electrodes, specific recording parameters (e.g., sampling rate, filtering range), and stimulation protocols used in the experiments. Provide the sources of antibodies, dilution factors, and specific experimental steps (e.g., incubation times and temperatures).

8-We have carefully revised the Methods section and integrated all the requested details. Specifically, we now describe in detail the environmental conditions for cell culture, including temperature, CO₂ concentration, and the exact composition of the culture media. For electrophysiological experiments, we added comprehensive information about electrode preparation, sampling rates, filtering ranges, and the stimulation protocols used. Additionally, we included precise details for NAD⁺/NADH determination, immunocytochemistry, and Western blotting protocols, such as antibody sources, catalog numbers, dilution factors, incubation times, and temperatures. All these additions have been highlighted in red in the revised manuscript to facilitate identification by the Reviewers and Editors.

Overall, the study is well-designed, rigorous in methodology, and innovative in its findings, providing important insights into the role of metabolic regulation in neuronal excitability.

The results hold potential clinical significance for developing new anti-epileptic therapies. It is recommended that the authors revise the manuscript according to the above suggestions before acceptance for publication. The revised manuscript will be more convincing and provide an important reference for research in related fields.

-We thank the Reviewer for the positive comments.

REFEREE #2:

Inhibiting glycolysis with 2-deoxyglucose (2DG) suppresses seizures acutely and on chronic treatment. Mechanism of anti-seizure action remains under investigation. The author studied the action of 20 mM 2DG applied to cultured neurons for up to 24 hours. They studied the role of REST/NRSF in the effects of glycolysis inhibition on hippocampal neuron activity. Treatment with 2-deoxy-D-glucose (2DG) decreased the NADH/NAD⁺ ratio, increased REST/NRSF expression, and promoted its nuclear translocation. The amplitude of evoked (eEPSCs) and miniature (mEPSCs) excitatory postsynaptic currents was reduced in a REST/NRSF-dependent manner. This effect was associated with a REST/NRSF-dependent reduction in the size of GluA2-positive puncta and a decrease in GluA2 expression without changes in the density of excitatory synapses.

-We thank the REviewer for the constructive criticisms and suggestions aimed at improving the quality of our work.

1) Recent studies in experimental animals have revealed that 2DG has cardiac toxicity doses above 60 mg/Kg. See (Terse PS, Joshi PS, Bordelon NR, Brys AM, Patton KM, Arndt TP, Sutula TP. 2-Deoxy-d-Glucose (2DG)-Induced Cardiac Toxicity in Rat: NT-proBNP and BNP as Potential Early Cardiac Safety Biomarkers. *Int J Toxicol.* 2016 May;35(3):284-93. doi: 10.1177/1091581815624397. Epub 2016 Feb 2. PMID: 26838190; PMCID: PMC4864115.). A dose

of 37.5 mg/Kg is safe. The authors need to address this issue. What is the dose equivalent of 20 mM 2 DG?)

1- We thank the Reviewer for raising the issue of 2DG toxicity and for citing the study by Terse et al. (*Int J Toxicol*, 2016), which demonstrated that chronic administration of 2DG above 60 mg/kg in rats can lead to cardiac toxicity. We fully acknowledge these findings and appreciate the opportunity to clarify this issue. First, no direct quantitative equivalence exists between in vitro concentrations (e.g., 20 mM) and in vivo doses (e.g., mg/kg). Pharmacokinetic profiles, metabolic rates, tissue distribution, and extracellular glucose levels differ markedly between in vitro and in vivo systems, limiting any straightforward extrapolation. For instance, standard neuronal culture media consistently contain 25 mM glucose (for reasons extensively discussed in our response to Reviewer 1, Comment 3), which far exceeds the physiological brain glucose levels (approximately 1–2 mM). In vitro, relatively high concentrations of 2DG are necessary to effectively inhibit glycolysis in this high-glucose environment. To improve physiological relevance, we tested lower glucose concentrations (maintaining a 1:2 glucose:2DG ratio). However, as reported in our response to Reviewer 1 (Comment 3), these conditions resulted in significant neuronal death, precluding their use in this study. Our dose–response viability assays (5 mM glucose/10 mM 2DG, 2 mM glucose/4 mM 2DG, and 1 mM glucose/2 mM 2DG) revealed that glucose concentrations below 10 mM, in combination with 2DG, caused substantial toxicity (Figure 1B). Consequently, the combination of 10 mM glucose with 20 mM 2DG was the only condition that preserved neuronal viability comparable to that observed under standard glucose levels (either 10 or 25 mM glucose alone), and this combination was therefore used throughout our experiments.

Regarding the in vivo toxicity, it is important to note that the “toxic dose” of 2DG is highly context-dependent, varying with species, strain, route of administration (oral, intraperitoneal, osmotic pump), treatment duration, age, and diet. Notably, several studies have shown that ketogenic diets or alternative metabolic substrates (e.g., β -hydroxybutyrate) can significantly improve tolerance to 2DG by mitigating systemic side effects (Lian et al., *Cell Metab*, 2007; Diano et al., *Cell Metab*, 2011). Thus, the 60 mg/kg threshold reported by Terse et al. reflects a specific chronic-dosing protocol in rats under a standard diet and should not be generalized across different experimental settings. In summary, our use of 20 mM 2DG in vitro is both safe and necessary to achieve effective glycolytic inhibition under controlled conditions of reduced glucose, as confirmed by both existing literature and our systematic toxicity testing. The use of 10 mM glucose already represents a compromise toward physiological relevance while preserving neuronal viability. Although a full in vitro dose–response study was beyond the scope of the present work, we recognize its importance and plan to explore it systematically in future studies. Finally, we note that the tolerability of 2DG has been established in several preclinical and clinical studies, with doses ranging from 60 to 250 mg/kg safely evaluated in oncology and neurology settings (Ockuly et al., *Epilepsy Res*, 2012; Sutula & Fountain, *Epilepsy Behav*, 2024). We have now included a dedicated paragraph in the revised Discussion to clarify this point and highlight the importance of careful, context-specific dose evaluation in future investigations

2) These studies are performed in cultured neurons with many autaptic connections. Therefore, these results may not apply in vivo.

2- We fully acknowledge that autaptic cultures cannot fully replicate the complexity of intact neuronal networks. However, only the electrophysiological experiments assessing evoked excitatory and inhibitory postsynaptic currents and their modulation by 2DG were performed in hippocampal neurons plated as autaptic cells.

The use of the autaptic model was driven by its unique experimental advantages. This preparation enables the isolation of a single excitatory/inhibitory neuron forming synapses onto itself, providing a highly controlled, cell-autonomous system to directly investigate the effects of 2DG on postsynaptic excitatory transmission. In this setup, all recorded synaptic inputs originate from the same neuron, eliminating confounding variability arising from network heterogeneity and allowing precise dissection of postsynaptic mechanisms affecting synaptic strength. All other experimental approaches in this study (including mEPSC recordings, immunocytochemistry, Western blotting,

and multielectrode array recordings) were conducted in primary hippocampal or cortical cultures at various cell densities, which better reflect the complexity of neuronal networks.

These diverse approaches allowed us to combine the mechanistic precision of autaptic recordings with the broader network context provided by mixed cultures. We have now clarified these methodological distinctions explicitly in the revised Materials and Methods section.

3) The authors argue that 2 DG has non-presynaptic effects on synaptic transmission. They interpreted their data to suggest that release probability was unchanged. However, this conclusion should be confirmed by measuring release probability by another method, e.g., the method of failures.

3- We thank the Reviewer for the valuable suggestion to assess synaptic release probability (Pr) using an independent method such as failure analysis. While we fully acknowledge the value of complementary approaches, we respectfully note that failure analysis is not suitable for our experimental system based on cultured hippocampal neurons. This method is most effective in acute brain slice preparations where minimal stimulation paradigms can be employed to activate a small number of synaptic contacts, yielding high and informative failure rates. In contrast, in cultured neurons, stimulation typically recruits multiple synapses, limiting the reliability of failure analysis. Moreover, a common strategy to increase failure rates—namely reducing extracellular Ca^{2+} concentrations—would interfere with the metabolic conditions we aim to preserve during 2DG treatment, introducing significant confounding factors into our experimental design.

To address the Reviewer's concern in a technically appropriate and rigorous manner, we employed an alternative and widely validated method to estimate both RRP and Pr: hyperosmotic sucrose stimulation (Rosenmund & Stevens, 1996). Low-density hippocampal neurons were voltage-clamped at -70 mV, and maximal extracellular stimulation was delivered via current pulses of variable amplitude using an isolated pulse stimulator, to evoke maximal eEPSCs by recruiting all excitatory afferents to the recorded neuron. One minute later, the same neuron was challenged with a hypertonic solution (500 mM sucrose in standard extracellular solution) applied using a rapid perfusion system. The charge transfer of the transient current induced by sucrose application was measured as the total readily releasable pool (RRP_{tot}). Pr was calculated as the ratio of the eEPSC charge to the RRP_{tot}. These additional data have been incorporated into Figure 6, panels D and E, to complement the previous findings. A comprehensive description of the methodological aspects of this approach has been incorporated into the Materials and Methods section.

As previously reported (Moulder & Mennerick, 2005), RRP_{tot} estimated via hyperosmotic sucrose application is typically 5–6 times larger than RRP_{syn} inferred from cumulative EPSC analysis during high-frequency stimulation. Consequently, Pr values calculated from sucrose-based estimates tend to appear lower. This discrepancy arises because sucrose induces a massive, Ca^{2+} -independent release of all docked and fusion-competent vesicles, including those not normally recruited by physiological stimuli. In contrast, cumulative analysis captures only the subset of vesicles that actively participate in Ca^{2+} -dependent, action potential-evoked release (RRP_{syn}), thereby underestimating the full size of the readily releasable pool.

Importantly, both the RRP_{syn} (estimated via cumulative EPSC analysis) and RRP_{tot} (estimated via sucrose stimulation) reflect the product of the number of readily releasable vesicles and the postsynaptic response to the content of a single vesicle (i.e., quantal size, corresponding to the mean mEPSC amplitude). In our experiments, 2DG treatment

significantly reduced mEPSC amplitude, indicating a reduction in quantal size. Thus, the observed decrease in both RRP_{syn} and RRP_{tot} under 2DG treatment is not attributable to a reduced number of readily releasable vesicles, but rather to a diminished postsynaptic response per vesicle.

This conclusion is supported by multiple converging lines of evidence:

- No change in Pr, as independently assessed by both cumulative EPSC analysis and sucrose stimulation;
- No change in paired-pulse ratio (PPR);
- No change in mEPSC frequency;
- Significant reduction in mEPSC amplitude;
- Decreased GluA2 expression, confirmed by Western blotting, immunocytochemistry, and supported by new pharmacological experiments using IEM-1460.

Taken together, these findings strongly support a REST/NRSF-mediated postsynaptic downscaling mechanism induced by 2DG, with no evidence of presynaptic alterations.

4. The authors claim that EPSC amplitude is reduced because of reduced GluA2 expression. GluA2 lacking receptors rectify and are sensitive to IEM and Philanthotoxin. Authors need to confirm their findings of diminished GluA2 expression with electrophysiology.

As correctly noted, GluA2-lacking AMPA receptors exhibit inward rectification and are selectively sensitive to polyamine-derived antagonists, such as IEM-1460 and Philanthotoxin-433. To assess this, we performed new electrophysiological recordings using IEM-1460 (100 μ M) to evaluate the contribution of GluA2-lacking AMPA receptors to evoked EPSCs in autaptic hippocampal neurons. In vehicle / NEG-treated neurons, IEM-1460 reduced EPSC amplitude by ~20%, consistent with the low proportion of GluA2-lacking AMPARs typically observed in mature (DIV14–16) excitatory neurons. In contrast, in NEG / 2DG neurons, the IEM-1460-induced block increased to ~50%, indicating a higher prevalence of GluA2-lacking receptors. These electrophysiological findings are consistent with our Western blot and immunocytochemistry data showing REST/NRSF-mediated downregulation of GluA2 (Figure 8E–G). The IEM-1460 experiments are now included in Figure 8 (panels H, I) and support the conclusion that 2DG promotes a REST/NRSF-driven shift in AMPA receptor subunit composition toward GluA2-lacking receptors. These results complement the molecular data and provide converging evidence that the observed EPSC reduction may reflect both quantitative and compositional changes in AMPA receptor expression.

Minor:

1) Please specify if the experimenter was blinded to 2 DG/ODN treatment.

1- All experiments reported in this manuscript—including imaging, electrophysiology, biochemical analyses (Western blot and qPCR), and multi-electrode array (MEA) recordings—were performed under blinded conditions. The blinding procedure was systematically applied to all experimental protocols to minimize bias and ensure the robustness and reproducibility of the findings. We now explicitly state it in the revised Methods section.

2) How was sampling bias addressed when selecting cells to study REST relocation to the nucleus?

2- We confirm that all imaging experiments assessing REST/NRSF nuclear translocation were performed under blinded conditions. To minimize sampling bias, cells were selected objectively based on a predefined fluorescence intensity threshold criterion. Specifically, only cells showing at least 20% coverage of the somatic area by the red ODN-Cy3 fluorescence signal were considered for analysis. This criterion ensures that only cells with adequate internalization of the fluorescent decoy probe were analyzed. No additional selection criteria were applied beyond this threshold, and cell selection within each imaging field was randomized among qualifying cells. We now clarify this important point in the Methods section of the revised manuscript.

3) Drugs used to treat epileptic seizures were called Anti epileptic drugs but are best referred to as Antiseizure medications.

3- We have carefully revised the manuscript and replaced the term “antiepileptic drugs” with the more appropriate term “antiseizure drugs/medications” throughout the entire text, in accordance with the current terminology.

Dear Professor Baldelli,

Re: JP-RP-2025-288763R1 "Harnessing Metabolic Control for Synaptic Stability: REST/NRSF Links Glycolytic Inhibition to Excitatory Neurotransmission" by Pietro Baldelli, Pierluigi Valente, Franco Onofri, Santina Bruzzzone, Fabio Benfenati, Daniele Ferrante, Barbara Parisi, Antonella Marte, Dionisi Marianna, Andrea Benzi, and Cosimo Prestigio

We are pleased to tell you that your paper has been accepted for publication in The Journal of Physiology.

- The Journal of Physiology funds authors of provisionally accepted papers to use the premium BioRender site to create high resolution schematic figures. Follow this link and enter your details and the manuscript number to create and download figures. Upload these as the figure files for your revised submission. If you choose not to take up this offer, we require figures to be of similar quality and resolution. If you are opting out of this service to authors, state this in the Comments section on the Detailed Information page of the submission form. The link provided should only be used for the purposes of this submission. Authors will be charged for figures created on this premium BioRender account if they are not related to this manuscript submission.

- You must upload original, uncropped western blot/gel images (including controls) if they are not included in the manuscript. This is to confirm that no inappropriate, unethical or misleading image manipulation has occurred. These should be uploaded as 'Supporting information for review process only'. Please label/highlight the original gels so that we can clearly see which sections/lanes have been used in the manuscript figures. For more information, see: <https://physoc.onlinelibrary.wiley.com/hub/journal-policies#magmanip>.

Yours sincerely,

Katalin Toth
Senior Editor
The Journal of Physiology

If you would like to receive our 'Research Roundup', a monthly newsletter highlighting the cutting-edge research published in The Physiological Society's family of journals (The Journal of Physiology, Experimental Physiology, Physiological Reports, The Journal of Nutritional Physiology and The Journal of Precision Medicine: Health and Disease), please click this link, fill in your name and email address and select 'Research Roundup': <https://www.physoc.org/journals-and-media/membernews>

- You can help your research get the attention it deserves! Check out Wiley's free Promotion Guide for best-practice recommendations for promoting your work at: www.wileyauthors.com/eoo/guide. You can learn more about Wiley Editing Services which offers professional video, design, and writing services to create shareable video abstracts, infographics, conference posters, lay summaries, and research news stories for your research at: www.wileyauthors.com/eoo/promotion.

EDITOR COMMENTS

Reviewing Editor:

Comments to the Author:

Thank you for submitting a revised manuscript. Both referees agree that their comments have been fully addressed.

REFEREE COMMENTS

Referee #1:

The authors have addressed all of my concerns in this revised version. The present study provides robust evidence supporting the novel "metabolism-epigenetics-synaptic function" mechanistic axis, offering new theoretical insights and potential therapeutic targets for epilepsy and other neurological disorders. I therefore recommend the manuscript for publication.

Referee #2:

The authors have addressed my concerns with additional experiments and explanations.

Reviewer comments

The authors have addressed all of my concerns in this revised version. The present study provides robust evidence supporting the novel “metabolism – epigenetics – synaptic function” mechanistic axis, offering new theoretical insights and potential therapeutic targets for epilepsy and other neurological disorders. I therefore recommend the manuscript for publication.